# lncRNA read-through regulates the BX-C insulator *Fub-1*

Airat Ibragimov[1,2]*, Xin Yang Bing[3†], Yulii V Shidlovskii[4,5], Michael Levine[3], Pavel Georgiev[6], Paul Schedl[1]*

[1]Department of Molecular Biology, Princeton University, Princeton, United States; [2]Center for Precision Genome Editing and Genetic Technologies for Biomedicine, Institute of Gene Biology, Russian Academy of Sciences, Moscow, Russian Federation; [3]Lewis Sigler Institute, Princeton University, Princeton, United States; [4]Laboratory of Gene Expression Regulation in Development, Institute of Gene Biology Russian Academy of Sciences, Moscow, Russian Federation; [5]Department of Biology and General Genetics, Sechenov University, Moscow, Russian Federation; [6]Department of the Control of Genetic Processes, Institute of Gene Biology Russian Academy of Sciences, Moscow, Russian Federation

**Abstract** Though long non-coding RNAs (lncRNAs) represent a substantial fraction of the Pol II transcripts in multicellular animals, only a few have known functions. Here we report that the blocking activity of the Bithorax complex (BX-C) *Fub-1* boundary is segmentally regulated by its own lncRNA. The *Fub-1* boundary is located between the *Ultrabithorax* (*Ubx*) gene and the *bxd/pbx* regulatory domain, which is responsible for regulating *Ubx* expression in parasegment PS6/segment A1. *Fub-1* consists of two hypersensitive sites, *HS1* and *HS2*. *HS1* is an insulator while *HS2* functions primarily as an lncRNA promoter. To activate *Ubx* expression in PS6/A1, enhancers in the *bxd/pbx* domain must be able to bypass *Fub-1* blocking activity. We show that the expression of the *Fub-1* lncRNAs in PS6/A1 from the *HS2* promoter inactivates *Fub-1* insulating activity. Inactivation is due to read-through as the *HS2* promoter must be directed toward *HS1* to disrupt blocking.

**\*For correspondence:**
airati@princeton.edu (AI);
pschedl@princeton.edu (PS)

**Present address:** †BlueRock Therapeutics, Cambridge, United States

## Editor's evaluation

This study provides compelling evidence for the mechanism of an insulator element, which establishes boundaries between gene neighborhoods to allow proper gene regulation. In the fruit fly *Drosophila melanogaster*, the bithorax complex contains a series of Hox genes that determines segment identity. The authors show that transcriptional read-through of an bithorax insulator controls its activity and is used for proper patterning of the embryo.

## Introduction

The vast majority of Pol II transcripts encoded by the genomes of multicellular animals correspond to long non-coding RNAs (lncRNAs), not mRNAs. Their wide distribution in the genome and the fact that their expression is often coordinated with genes in the immediate vicinity have led to the idea that they have functions in gene regulation and chromosome organization (*Herman et al., 2022*; *Li and Fu, 2019*; *Núñez-Martínez and Recillas-Targa, 2022*; *Statello et al., 2021*). Unlike the sequences of protein coding mRNAs, which are typically conserved across species, the primary sequences of lncRNAs are usually not very well conserved. However, some lncRNAs contain short sequence blocks that exhibit a relatively high degree of conservation, even though most of the lncRNA are divergent. In these instances, the lncRNA itself has regulatory activities. A classic example is the mammalian X

chromosome inactivation lncRNA Xist. Conserved RNA sequence blocks within the Xist transcript function in recruiting Polycomb (PcG) silencing factors and targeting the transcript to the X chromosome (*Jacobson et al., 2022*; *Lu et al., 2020*; *Pandya-Jones et al., 2020*). Other lncRNAs, like the HoxA complex lncRNA *HOTT1P*, encode RNA sequences that recruit chromatin modifiers that help promote gene expression (*Wang et al., 2011*). In other cases, the lncRNA itself does not appear to have important functions. Instead, it is the regulatory elements (enhancers, silencers, and promoters) that control the expression of the lncRNA that are functionally important (*Ibragimov et al., 2020*). For example, the expression of the Myc gene is downregulated by a mechanism in which the promoter for the *PVT1* lncRNA competes with Myc for physical interactions with a set of shared enhancers. In other cases, the transcription of the lncRNA appears to be the important regulatory function. Isoda et al. found that transcription of *ThymoD* lncRNA helps regulate the expression of Bcl11b, a gene that plays an important role in specifying T cell fate (*Isoda et al., 2017*). Transcription of the lncRNA induces the demethylation of recognition sites for the chromosomal architectural protein CTCF, enabling CTCF to bind to these sites and induce looping between the Bcl11b enhancers and the Bcl11b gene.

In the studies reported here, we have investigated the role of lncRNAs in regulating the chromatin organization and expression of the homeotic gene, *Ultrabithorax* (*Ubx*), in the *Drosophila* bithorax complex (BX-C). *Ubx*, together with *abdominal-A* (*abd-a*) and *Abdominal-B* (*Abd-B*), are responsible for specifying the identity of the nine parasegments (PS)/segments that form the posterior two-thirds of the fly (*Maeda and Karch, 2015*: *Figure 1*). Parasegment identity depends upon a series of nine regulatory domains that direct the temporal and tissue-specific patterns of expression of the three BX-C homeotic genes. *Ubx* is responsible for specifying parasegments PS5 (adult cuticle segment T3) and PS6 (adult cuticle segment A1). Its expression in these two parasegments is controlled by the *abx/bx* and *bxd/pbx* regulatory domains, respectively. *abd-A* expression in PS7-PS9 is directed by the *iab-2–iab-4* regulatory domains, while *Abd-B* expression in PS10-14 is regulated by the *iab-5–iab-9* regulatory domains. Each regulatory domain has an initiation element, a set of tissue-specific enhancers, and Polycomb Response Elements (PREs) (*Iampietro et al., 2010*; *Maeda and Karch, 2015*; *Maeda and Karch, 2006*). Early in development, a combination of maternal, gap, and pair-rule gene proteins interact with the initiation elements in each regulatory domain, and set the domain in either the '*off*' or '*on*' state. The BX-C regulatory domains are sequentially activated along the anterior–posterior axis of the embryo. For example, in PS5(T3) only the *abx/bx* domain is activated (all of the remaining BX-C domains including *bxd/pbx* are '*off*') and it is responsible for regulating *Ubx* expression in this parasegment. Both *abx/bx* and *bxd/pbx* are turned on in PS6(A1); however, *Ubx* expression and parasegment identity depend on the *bxd/pbx* domain. Once the activity state of the domain is set in early embryos, it is remembered during the remainder of development by the action of Polycomb Group proteins (PcG: *off*) and Trithorax group proteins (Trx: *on*). Each domain also contains tissue- and stage-specific enhancers that drive the expression of its target homeotic gene in a pattern appropriate for proper development of the parasegment it specifies.

In order to properly specify parasegment identity, the nine BX-C regulatory domains must be functionally autonomous. Autonomy is conferred by boundary elements (insulators) that flank each domain (*Maeda and Karch, 2015*). The role of boundaries in BX-C regulation is best understood in the *Abd-B* region of the complex. Mutations that inactivate *Abd-B* boundaries cause a gain-of-function (GOF) transformation in segment identity. For example, the *Fab-7* boundary separates the *iab-6* and *iab-7* regulatory domains that are responsible for specifying PS11(A6) and PS12 (A7) identity, respectively (*Gyurkovics et al., 1990*). When *Fab-7* is deleted, the *iab-6* initiator activates the *iab-7* domain in PS11(A6) and *iab-7* instead of *iab-6* drives *Abd-B* expression in this parasegment as well as in PS12(A7). This results in the duplication of the PS12/A7 segment. However, blocking crosstalk between adjacent regulatory domains is not the only function of BX-C boundaries. The three homeotic genes and their associated regulatory domains are arranged so that all but three of the domains are separated from their targets by one or more intervening boundaries. For this reason, all but two (*Fub* and *Mcp*) of the internal BX-C boundaries must also have bypass activity (*Hogga et al., 2001*; *Kyrchanova et al., 2019a*; *Kyrchanova et al., 2023*; *Kyrchanova et al., 2019b*; *Kyrchanova et al., 2015*). For example, *Fab-7* has bypass activity and it mediates regulatory interactions between *iab-6* and *Abd-B*. However, when *Fab-7* is replaced by generic fly boundaries (e.g., *scs* or *su(Hw)*) or by BX-C boundaries that lack bypass activity (*Mcp*), these boundaries block crosstalk but do not support bypass (*Hogga et al.,*

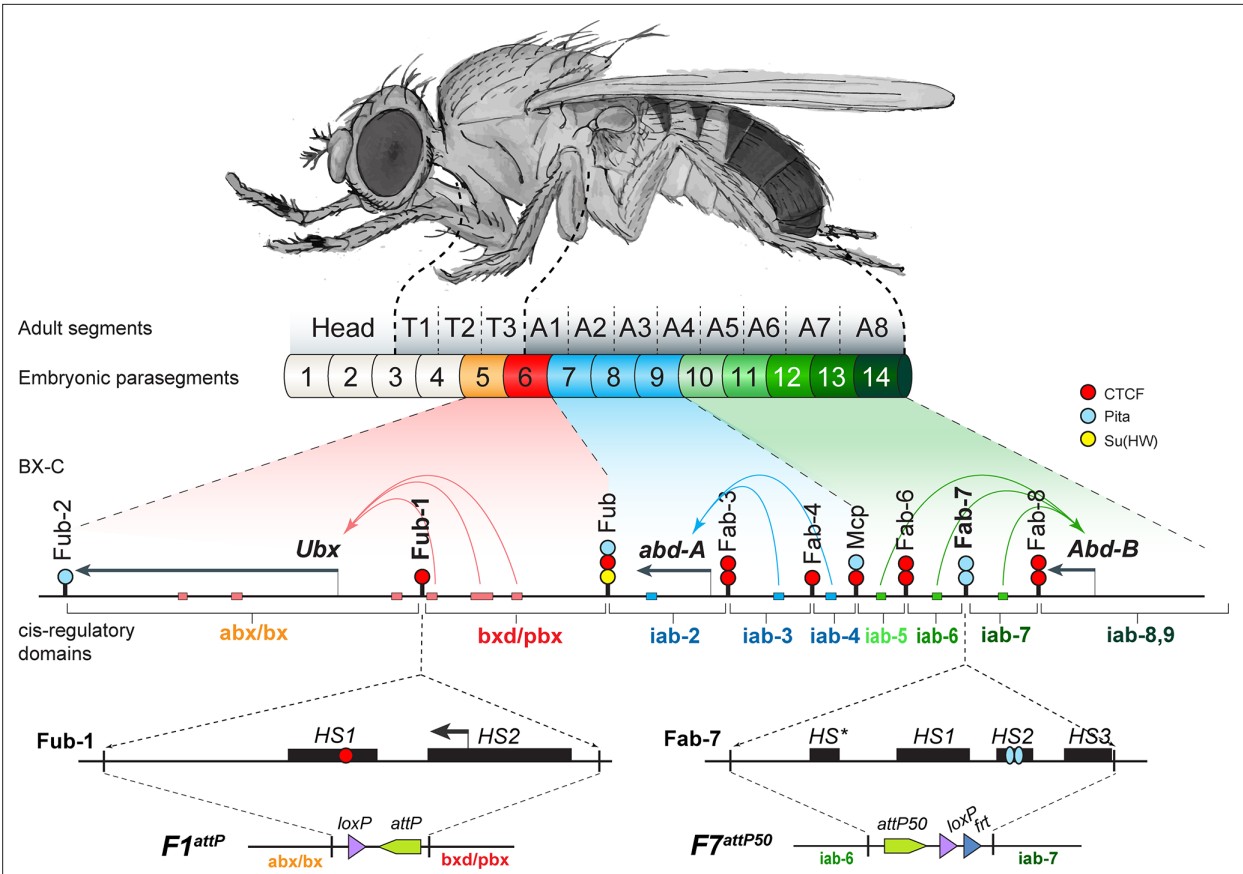

**Figure 1.** The organization of the genes and regulatory domains in BX-C. The *Drosophila melanogaster* Bithorax complex (BX-C), which contains the Hox genes *Ubx*, *abd-A*, and *Abd-B*, is shown in relation to where these three genes are expressed in an embryo. There are nine *cis*-regulatory domains that are responsible for the regulation of *Ubx* (*abx/bx* and *bxd/pbx* domains), *abd-A* (*iab-2*–*4* domains), and *Abd-B* (*iab-5*–*7* and *iab-8,9* domains), and for the development of parasegments 5–13 (PS)/segments (T3–A8). The anterior limit of expression of the three Hox gene is indicated by color coding: red: *Ubx*; blue: *abd-A*; green: *Abd-B* (reviewed in **Maeda and Karch, 2015**). The lines with colored circles mark chromatin boundaries. dCTCF, Pita, and Su(Hw) binding sites at the boundaries are shown as red, blue, and yellow circles/ovals, respectively. Embryonic enhancers are indicated by pink, blue, and green bars on coordinate line. On the bottom of the figure, the molecular maps of the *Fub-1* and *Fab-7* boundaries are shown, including their deletions. Transposase/nuclease hypersensitive sites are shown as black boxes above the coordinate bar. The proximal and distal deficiency endpoints of the *Fub-1* and *Fab-7* deletions used in the replacement experiments are indicated by vertical lines. The *attP*, *lox*, and *frt* sites used in genome manipulations are shown as green, violet, and blue triangles, respectively.

The online version of this article includes the following figure supplement(s) for figure 1:

**Figure supplement 1.** Chip-seq profiles of different chromatin proteins and histone modifications across *bxd/pbx* domain.

**Figure supplement 2.** *Fub-1* sequence conservation.

**Figure supplement 3.** Strategy for creating *Fub-1* deletion and replacement lines.

*2001*; **Kyrchanova et al., 2017**). As a consequence, *Abd-B* expression in PS11/A6 is driven by *iab-5* not *iab-6*, and this results in a loss-of-function (LOF) phenotype.

The *Ubx* domain is delimited and subdivided by three boundaries called *Fub-2*, *Fub-1*, and *Fub*. *Fub-2* and *Fub-1* flank *abx/bx* domain, whereas *bxd/pbx* is flanked by *Fub-1* and *Fub* (**Figure 1**; **Bowman et al., 2014**; **Mateo et al., 2019**). As *Fub-1* separates enhancers in *bxd/pbx* regulatory domain from *Ubx* gene (**Figure 1**), it would have to be bypassed so that *bxd/pbx* can direct *Ubx* expression in PS6/A1. Unlike the *Abd-B* region of the complex where multiple boundaries are located between the *iab-5* and the *Abd-B* promoter regulatory domain, there is only one boundary between *bxd/pbx* and the *Ubx* gene. As a consequence, the bypass mechanism might differ from that deployed elsewhere in BX-C. Here we show that segmentally regulated transcription of an lncRNA disrupts the blocking activity of *Fub-1*, enabling enhancers in the *bxd/pbx* regulatory domain to activate *Ubx* in PS6/A1 cells.

## Results

### The *bxd/pbx* regulatory domain: Boundaries and lncRNAs

The *bxd/pbx* regulatory domain controls *Ubx* expression in A1/PS6. This domain is *off* in more anterior parasegments (segments), while it is turned on in A1/PS6 and more posterior parasegments. The *bxd/pbx* domain is bracketed by the *Fub* boundary on the centromere distal side and *Fub-1* on the centromere proximal side. The chromosomal architectural proteins CTCF, Pita, and Su(Hw) are associated with *Fub* in ChIP experiments, and like boundaries the *Abd-B* region of BX-C *Fub* deletions result in a GOF transformation of PS6(A1) into PS7(A2). The *Fub-1* boundary has only been defined molecularly. ATAC-seq mapping in nuclear cycle 14 embryos indicates that this boundary includes two chromatin-specific transposase hypersensitive regions: *HS1* ~200 bp and *HS2* ~300 bp (*Figure 1*; *Hannon et al., 2017*). In ChIP experiments, *Fub-1* is bound by the chromosomal architectural proteins CTCF, CP190, Zw5, Pita, and Elba and marks the border of a Polycomb histone H3K27me3 domain in PS5/T3 cells (*Bowman et al., 2014*, p. 27; *Fuda et al., 2015*; *Harrison et al., 2011*; *Ueberschär et al., 2019*; *Zolotarev et al., 2016*; *Figure 1—figure supplement 1*). In addition, RNA Pol II and the histone mark for active transcription, H3K4me3, both map to *Fub-1*. Consistent with the results described below, the signal for H3K4me3 in PS7 cell ChIPs is greater than that observed in mixed ChIPs (*Bowman et al., 2014*, p. 27). In between *Fub-1* and the *Ubx* promoter, there are two prominent Zelda peaks and a putative enhancer (*Figure 1—figure supplement 1*; *Harrison et al., 2011*).

Besides directing the expression of *Ubx* in PS6(A1) and more posterior parasegments, the *bxd/pbx* domain also controls the expression of two lncRNAs, *bxd* and *Fub-1 HS2, Fub-1^{HS2}*. Pease et al. found that the *bxd* lncRNA initiates from a promoter close to the *pbx* enhancer and extends into the region located in between *Fub-1* and the *Ubx* promoter (*Pease et al., 2013*). The *bxd* lncRNA is first detected at the blastoderm stage as a broad band that extends from PS6/A1 to PS12/A7. Between the onset of gastrulation and stage 10 *bxd* lncRNA expression resolves into a series of eight stripes of differing intensity that span PS6/A1–PS12/A7. Pease et al. also detected a second *bxd/pbx*-dependent lncRNA, *Fub-1^{HS2}*. Like *bxd* lncRNA, it is expressed in PS6 (A1) and more posterior parasegments. This lncRNA is encoded by sequences located between *Ubx* and *Fub-1*, including sequences in *Fub-1 HS1*, and based on the in situ experiments of Pease et al., it likely originates from *Fub-1 HS2*. *Fub-1^{HS2}* shares exon sequences with the *bxd* lncRNA. However, the *Fub-1^{HS2}* lncRNA is detected in PS6/A1 and more posterior segments after the *bxd* lncRNA disappears.

As has been reported for the regulatory elements associated with other lncRNAs, the *Fub-1* hypersensitive sites are evolutionarily conserved. As shown in *Figure 1—figure supplement 2*, key sequence blocks within *HS1* and *HS2* are detected not only in other members of the *Sophophora* subgenus, but also in *Drosophila virilis,* a distantly related member of the *Drosophila* subgenus. The conserved blocks in *HS1* include the dCTCF recognition site, while in *HS2* the conserved blocks include several GAGA motifs that correspond to binding sites for the GAGA factor (GAF) as well as the predicted transcription start sites for the *Fub-1^{HS2}* lncRNA.

### *Fub-1* subdivides the *Ubx* regulatory domains

These observations, together with the studies of Bowman et al., showed that *Fub-1* marks the border of a Polycomb histone H3K27me3 domain in PS5 cells would predict that *Fub-1* delimits the endpoint of two adjacent TADs. The centromere distal TAD would encompass the *bxd/pbx* regulatory domain and end at *Fub* boundary that is located downstream of the *abd-A* transcription unit (*Bender and Lucas, 2013*; *Bowman et al., 2014*). The centromere proximal TAD would encompass the entire *Ubx* transcription unit and end at a predicted Pita/CP190 boundary, *Fub-2*, that is located between the 3′ end of the *Ubx* gene and the 5′ end of *modSP*.

To test these predictions, we used Micro-C to examine the TAD organization of *Ubx* and its two regulatory domains in 12–18 hr embryos. As shown in *Figure 2*, both regulatory domains (plus the *Ubx* transcription unit) are encompassed in a large TAD that extends from *Fub-2* on the centromere proximal side of *Ubx* to *Fub* on the distal side. Within this large TAD, there is an ~50 kb sub-TAD that spans *bxd/pbx* and is delimited by *Fub-1* on the left and *Fub* on the right (*Figure 2B and C*, black arrow). This *bxd/pbx* sub-TAD includes several smaller chromosomal segments that exhibit a high density of internal contacts (HDICs). The endpoint for one of these HDIC domains maps to *Fub-1*, while the other endpoint maps to the *bxd* PRE. There is a second HDIC domain linking *Fub-1* to sequences just upstream of the *Ubx* promoter. The *Ubx* promoter is also linked to the *bxd* PRE by a lower density

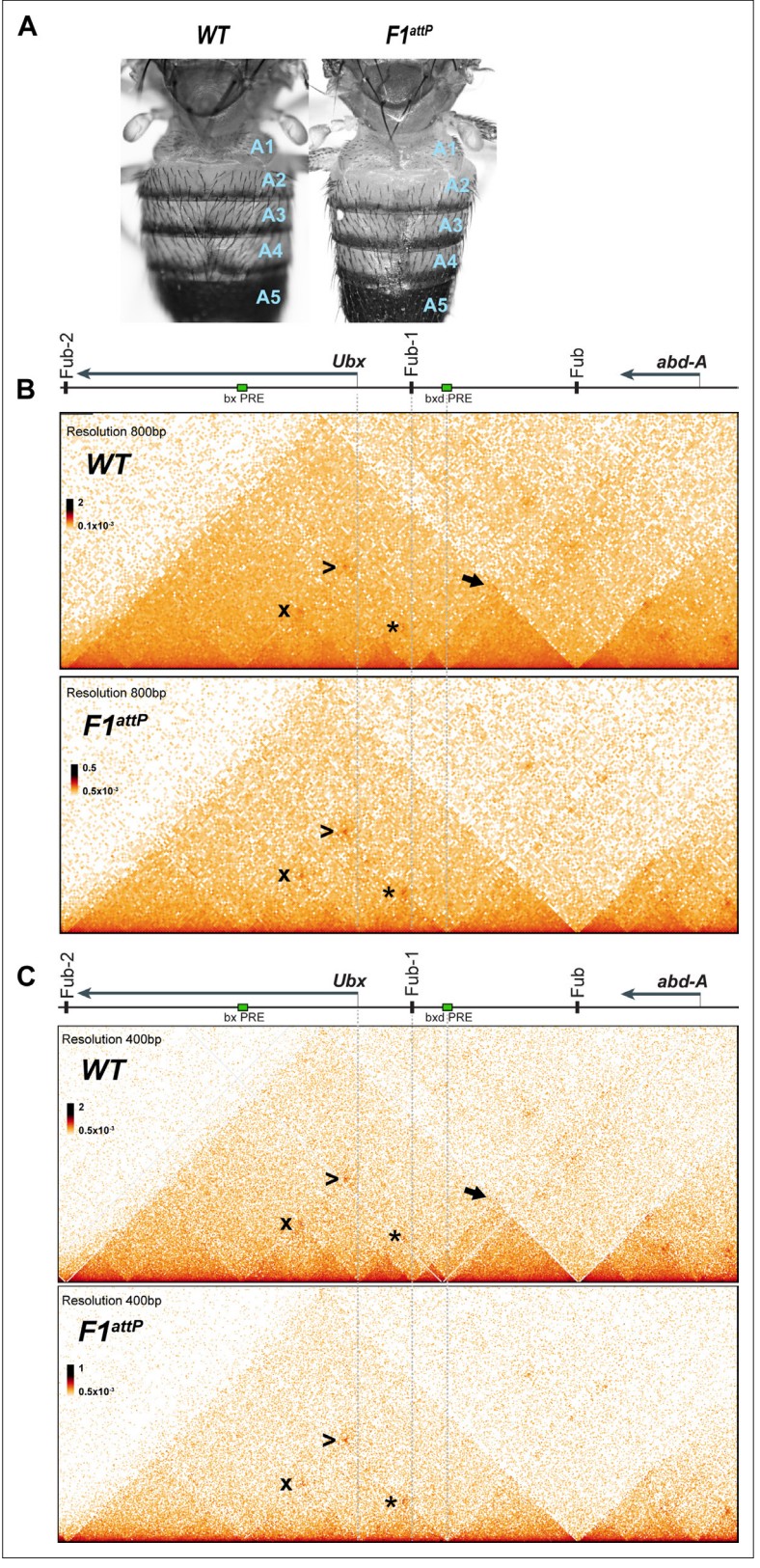

**Figure 2.** Chromatin topology in *bxd/pbx* domain in *WT* and in *Fub-1* deletion. (**A**) Morphology of the abdominal segments of *WT* and *F1^attP^* flies. (**B**) Micro-C contact map of the *WT* and *F1^attP^* 12–18 hr embryos at 800 bp resolution. The black arrow points to a sub-TAD formed by *Fub-1* and *Fub* boundaries. The *Ubx* promoter is linked to the *bxd* PRE by a lower density of internal contacts domain (LDIC) that is marked at the apex by an interaction

*Figure 2 continued on next page*

*Figure 2 continued*

dot (*). The *bxd* PRE forms an interaction dot (>) with the *bx* PRE as does the *Ubx* promoter (x). (**C**) Micro-C contact map of the *WT* and *F1^attP^* 12–18 hr embryos at 400 bp resolution.

of internal contacts domain (LDIC) that is marked at the apex by an interaction dot (*). The *bxd* PRE also forms an interaction dot (>) with the *bx* PRE as does the *Ubx* promoter (x). Interestingly PREs in other developmental loci have recently been shown to interact with each other and also function as 'tethering' elements helping to link distant enhancers to their target genes (*Batut et al., 2022*; *Eagen et al., 2017*; *Ibragimov et al., 2022*; *Levo et al., 2022*).

## *Fub-1* is not required to block crosstalk between *abx/bx* and *bxd/pbx*

A BX-C boundary in the location of *Fub-1* would be expected to have two regulatory functions. One would be blocking crosstalk between *abx/bx* and *bxd/pbx* while the other would be bypass so that enhancers in the *bxd/pbx* domain would be able to drive *Ubx* expression in PS6/A1. To test for blocking activity, we generated a 1168 bp deletion (*F1^attP^*) that retains an *attP* site for boundary replacement experiments (*Figure 1*, *Figure 1—figure supplement 3*). Deletions of boundaries elsewhere in BX-C typically result in GOF transformations. This transformation arises because initiators in the domain centromere proximal (*Figure 1*, right) to the boundary are able to active the domain distal to the boundary (*Figure 1*, left) in a parasegment in which the distal domain would normally be silent (*Barges et al., 2000*; *Bender and Lucas, 2013*; *Gyurkovics et al., 1990*; *Karch et al., 1994*; *Postika et al., 2021*). For this reason, we anticipated that the *bxd/pbx* domain would be inappropriately activated by initiators in *abx/bx* in PS5, and in adults we would observe a GOF transformation of T3 (PS5) toward A1 (PS6). Unexpectedly, however, *F1^attP^* flies do not show any evidence of a GOF transformation and their morphology is indistinguishable from wild type (*WT*) (*Figure 2A*).

A plausible explanation for this result is that there is a nearby element that can substitute for *Fub-1* and block crosstalk between *abx/bx* and *bxd/pbx*. *Figure 2B and C* show that the sub-TAD linking *Fub-1* to *Fub* that encompassed the *bxd/pbx* regulatory domain disappears in the *Fub-1* deletion. However, a new sub-TAD that includes *Fub* does not appear to be formed. Instead, the element located just upstream of the *Ubx* promoter forms a sub-TAD domain with the *bxd* PRE (see interaction dot in *Figure 2B and C* [*] at the apex). It may be that this topological configuration is sufficient to block crosstalk between initiators in *abx/bx* and *bxd/pbx* when the activity state of these domains in T3 (PS5) is set during early embryogenesis.

## The *Fub-1* element does not block enhancers in *bxd/pbx* from activating reporters

Although *Fub-1* defines one endpoint of a TAD and marks the border for the PcG H3K27me3 histone mark in PS5 nuclei, the finding that *Fub-1* deletions have no phenotype raises the possibility that it is actually 'within' the *bxd/pbx* regulatory domain. If this is the case, then reporters inserted between *Fub-1* and the element just upstream of the *Ubx* promoter should respond to *bxd/pbx* enhancers, but be insulated from enhancers in the *abx/bx* regulatory domain. To test this possibility, we analyzed the pattern of expression of reporters inserted into the *F1^attP^* site with and without the *Fub-1* boundary.

We first examined the *DsRed* reporter used to mark the *Fub-1* deletion in *DsRed-F1^attP^*. *Figure 3* shows that *DsRed* is expressed throughout the posterior ~2/3rds of *Fub-1* deletion embryos, with an anterior limit corresponding to the anterior border of PS6/A1. Thus, as expected from the lack of phenotypic effects of the *Fub-1* deletion, the reporter is insulated from the *abx/bx* domain, but is subject to regulation by the *bxd/pbx* domain. We next introduced the 1168 bp *Fub-1* fragment together with a *GFP* reporter flanked by *frt* sites back into the *DsRed-F1^attP^* platform (*Figure 1—figure supplement 3*). In the resulting insert, the *Fub-1* element is flanked by *DsRed* on the proximal *Ubx* side and *GFP* on the distal *bxd/pbx* side. The reporters were then excised individually to give *DsRed-Fub-1* and *Fub-1-GFP* (*Figure 1—figure supplement 3*). As shown in *Figure 3*, the anterior border of expression of both *DsRed* and *GFP* corresponds to the PS6/A1 as is observed for *DsRed* when the *Fub-1* element is deleted. Thus *Fub-1* is unable to block the *bxd/pbx* enhancers from activating the *DsRed* reporter in PS6/A1 cells (and more posterior parasegments/segments), and this reporter is also 'located' within the *bxd/pbx* regulatory domain.

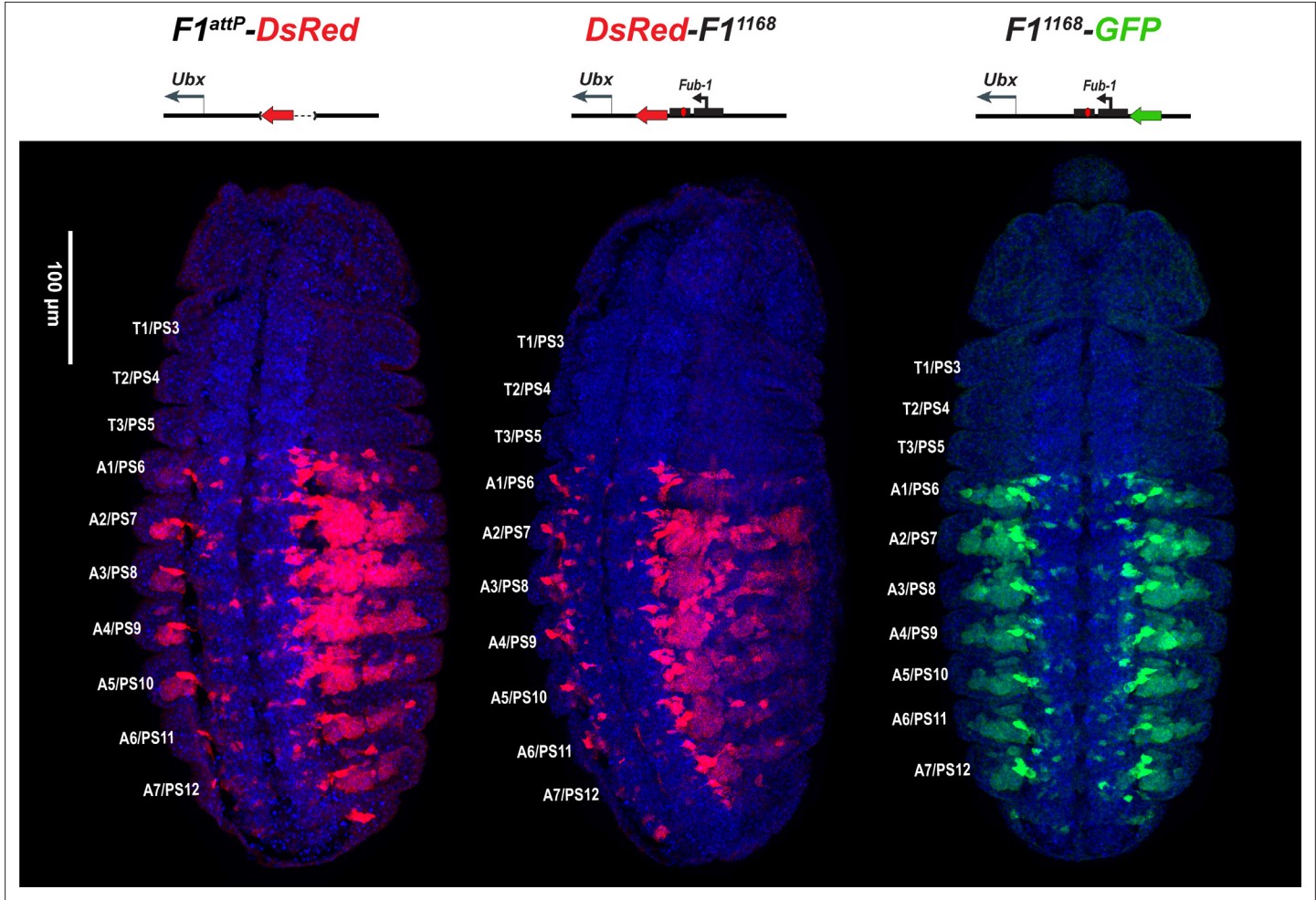

**Figure 3.** *DsRed* and *GFP* markers trap enhancer activity in *Fub-1* replacements. *DsRed* (in red) and *GFP* (in green) expression in stage 14 embryos in the *Fub-1* deletion and in two replacements as indicated. DAPI was used to stain nuclei (in blue). *DsRed* expression in *F1^attP^-DsRed* begins in PS6/A1. In *DsRed-F1^1168^* and *F1^1168^-GFP*, expression patterns of both markers are limited to PS6/A1-PS13/A8, consistent with the idea that the *Ubx* promoter region can function to demarcate the *abx/bx* and *bxd/pbx* domains.

These findings fit with previous enhancer trap studies that showed that P-element insertions around the *Ubx* promoter displayed two distinct patterns of expression. An enhancer trap located 13 bp upstream of the *Ubx* transcription start site had a PS5/T3 anterior border, indicating that it is regulated by the *abx/bx* domain. In contrast, an enhancer trap located 196 bp upstream of the start site had an anterior limit of expression shifted to PS6/A1 (*McCall et al., 1994*). Taken together with the experiments described in the previous section, it would appear that the *Ubx* promoter region is sufficient for the functional autonomy of the *abx/bx* and *bxd/pbx* regulatory domains under laboratory growth conditions.

## Full-length *Fub-1* replacements of *Fab-7* do not block crosstalk between *iab-6* and *iab-7*

The results described above are unexpected for several reasons. First, *Fub-1* defines the proximal endpoint of a TAD that extends to *Fub* and encompasses the *bxd/pbx* regulatory domain. Second, it also corresponds to the proximal border of the PcG H3K27me3 histone mark in PS5 nuclei where *abx/bx* should be 'on' and *bxd/pbx* should be 'off.' Third, both of the *Fub-1* sub-elements, *HS1* and *HS2*, are evolutionarily conserved. In spite of these properties, deletion of *Fub-1* does not result in a detectable GOF transformation of T3/PS5 into A1/PS6. This is likely explained by the presence of the boundary element upstream of the *Ubx* promoter, which is able to block crosstalk between initiators in *abx/bx* and *pbx/bxd*. However, this would not explain why *Fub-1* fails to block the *pbx/bxd* enhancers

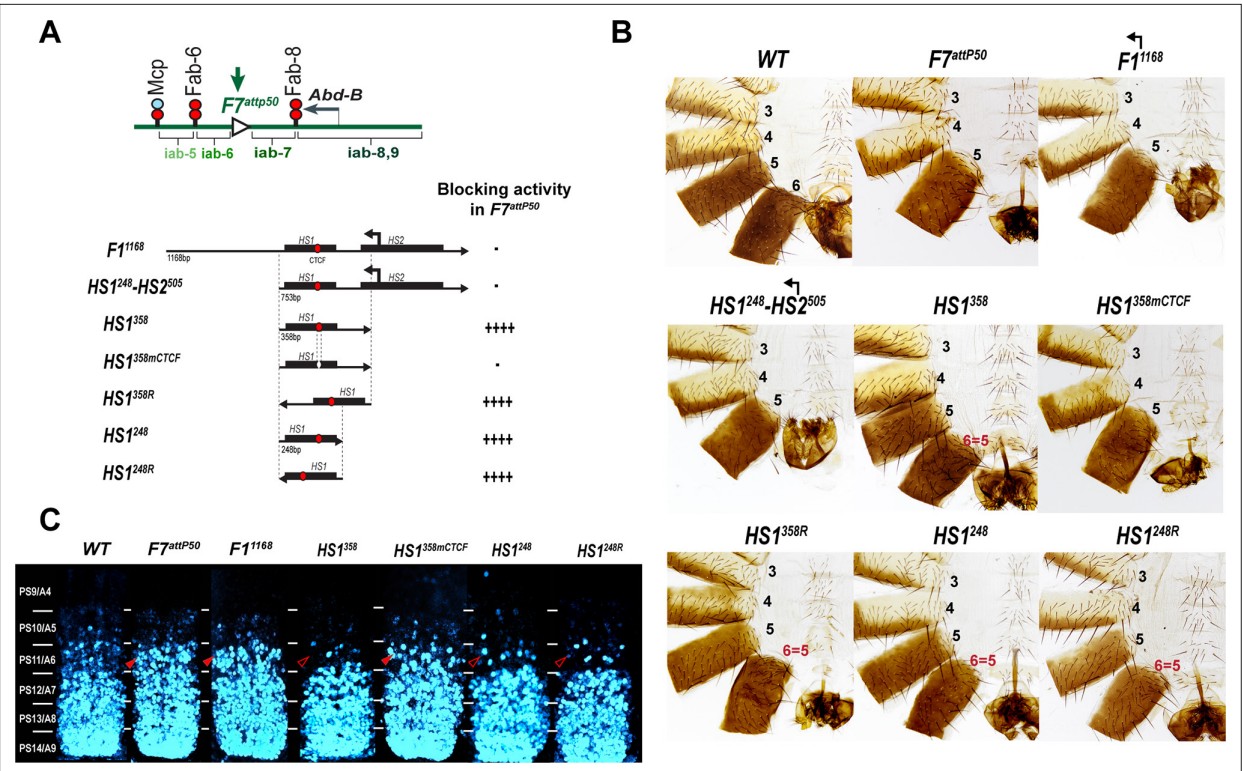

**Figure 4.** Testing boundary activity of *Fub-1* sequences. (**A**) Top: schematic presentation of *Fab-7* substitution. Bottom: *Fub-1* fragments used in the replacement experiments. On the right side, the insulator activity with the various fragments in the *F7^{attP50}* insertion site in adults as judged from cuticle preps. The number of '+' signs reflects the strength of insulator activity, where '++++' is full blocking, and '-' is lack of detectable blocking activity, respectively. Designations are the same as described in *Figure 1*. (**B**) Morphology of the male abdominal segments (numbered) in *F1^{1168}*, *HS1^{248}-HS2^{505}*, *HS1^{358}*, *HS1^{358mCTCF}*, *HS1^{358R}*, *HS1^{248}*, and *HS1^{248R}* replacements. (**C**) *Abd-B* expression in *Fab-7* replacement embryos. Each panel shows a confocal image of the embryonic CNS of stage 15 embryos stained with antibodies against *Abd-B* (cyan). The filled red arrowheads show morphological features indicative of gain-of-function (GOF) transformations. The empty red arrowheads show the signs of the loss-of-function (LOF) transformation, which is directly correlated with the boundary function of tested DNA fragments. The *WT* expression pattern of *Abd-B* in the embryonic CNS is characterized by a stepwise gradient of increasing protein level from PS10/A5 to PS14/A8. In *F7^{attP50}* embryos, *Abd-B* expression level in PS11/A6 is roughly equal to that in PS12/A7, indicating that *iab-7* drives *Abd-B* expression in PS11/A6 (GOF phenotype). Consistent with the adult phenotype, in *F1^{1168}* and *HS1^{248}-HS2^{505}* *Abd-B* expression in PS11/A6 is the same as in *F7^{attP50}*. The *Abd-B* expression pattern in *HS1^{358}* and *HS1^{248}* replacement is also consistent with the adult cuticular phenotypes: *Abd-B* expression is reduced in both PS10/A5 and PS11/A6 (LOF phenotype) compared with *WT*. In contrast, mutation of dCTCF site in *HS1^{358mCTCF}* results in the loss of blocking activity and *Abd-B* expression pattern similar to *F7^{attP50}*.

The online version of this article includes the following figure supplement(s) for figure 4:

**Figure supplement 1.** Morphology of the abdominal segments (numbered) in males carrying different variants of *Fub-1* in *Fab-7^{attP50}* platform in the dark field.

**Figure supplement 2.** Morphology of the abdominal segments (numbered) in males carrying *HS2^{505}* and *HS2^{505R}* fragments in *Fab-7^{attP50}*.

from activating a *DsRed* reporter when interposed between the enhancers and the reporter. These observations raise a novel possibility: even though *Fub-1* has many of the characteristic chromosome architectural functions of boundary elements, it differs from other boundaries in flies and other organisms in that it lacks insulating activity.

To test whether *Fub-1* lacks insulating activity, we took advantage of the *Fab-7^{attP50}* (*F7^{attP50}*) replacement platform (*Figure 1*; *Wolle et al., 2015*). In this platform, the *Fab-7* boundary was deleted, and an *attP* site introduced in its place. The deletion of *Fab-7* fuses the *iab-6* and *iab-7* regulatory domains, enabling parasegment-specific initiation elements in *iab-6* to ectopically activate *iab-7* (*Karch et al., 1994*; *Wolle et al., 2015*). As a consequence, *iab-7* drives *Abd-B* expression in PS11, transforming PS11/A6 into a duplicate copy of PS12/A7 (*Figure 4B and C*). The *attP* site in the *F7^{attP50}* can be used to insert any sequences of interest to test for two different boundary functions. The first is blocking crosstalk between *iab-6* and *iab-7*, while the second is supporting bypass so that *iab-6* can regulate *Abd-B* in PS11/A6.

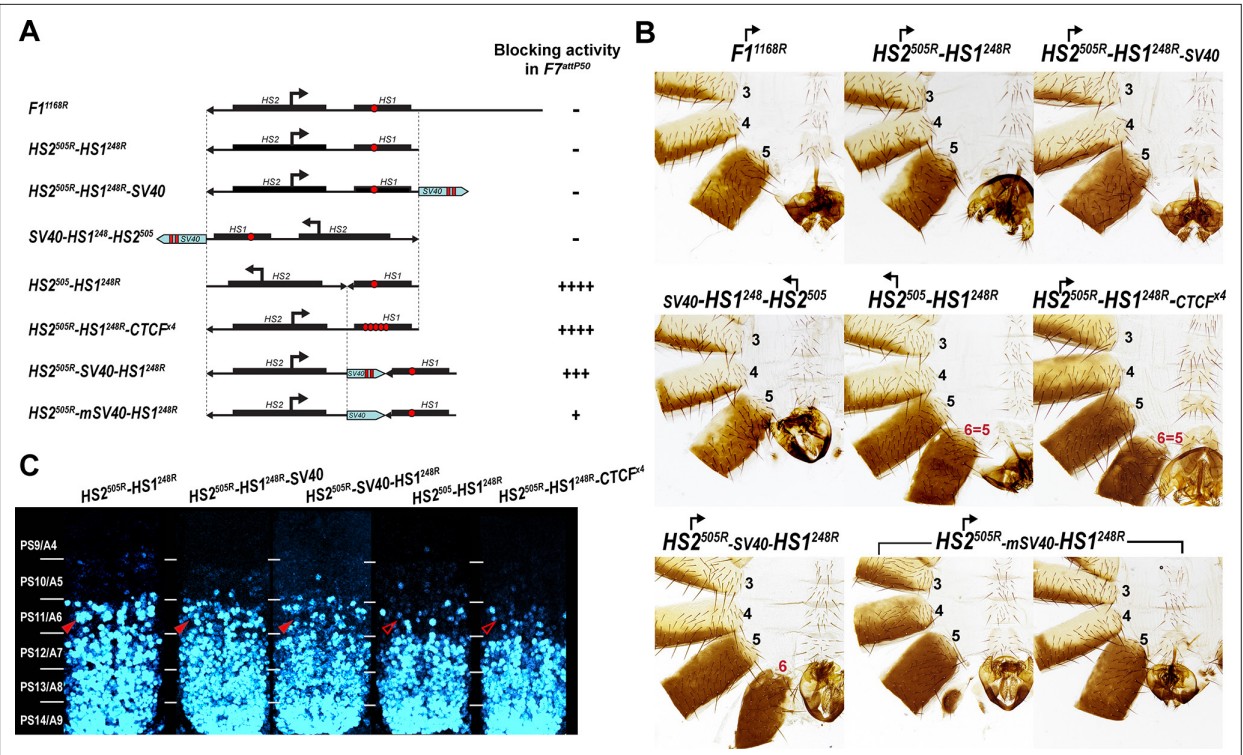

**Figure 5.** Read-through transcription inactivates *HS1* insulator. (**A**) Schematic representation of sequences tested in *F7^attP50*. SV40 terminator is shown as a light blue arrow, and poly(A) sites are marked as vertical red lines. All other designations are the same as described in **Figure 4**. (**B**) Morphology of the male abdominal segments (numbered) of *Fab-7* replacements. (**C**) *Abd-B* expression in CNS of *Fab-7* replacement embryos. Embryos were stained and marked as in **Figure 4**.

The online version of this article includes the following figure supplement(s) for figure 5:

**Figure supplement 1.** *Abd-B* expression in *Fab-7* replacement embryos.

In the first experiment, we used the *F7^attP50* replacement platform to test the blocking activity of a 1168 bp sequence spanning the entire deletion, *Fub-1^1168*, and a 753 bp sequence including only hypersensitive sites, *HS1^248-HS2^505* (R6.22: 3R:16748578–16749330) inserted in the same 5′→3′ orientation as the endogenous *Fub-1* element. **Figure 4B** and **Figure 4—figure supplement 1** show that males carrying either the 1168 bp or the 753 bp replacements lack A6 just like the starting *F7^attP50* deletion. This result indicates that these two *Fub-1* fragments do not prevent crosstalk between *iab-6* and *iab-7*. The *HS1^248-HS2^505* replacement GOF phenotype is reflected in the pattern of *Abd-B* expression in the embryonic CNS that closely matches that of the starting *F7^attP50* deletion (**Figure 4C**). We also inserted the two *Fub-1* fragments in the reverse orientation; however, in this case also neither of the fragments rescued the GOF phenotype of the *F7^attP50* deletion (**Figure 5**).

## *HS1* has boundary activity

Previous replacement experiments have shown that dCTCF-associated BX-C boundaries like *Mcp* are able to block crosstalk between *iab-6* and *iab-7* (**Postika et al., 2018**). For this reason, we wondered whether *HS1* alone has blocking activity. We used two different *HS1* fragments to test this possibility. One was 358 bp (R6.22: 3R: 16748578–16748935) and included all of the *HS1* sequences plus the sequences located between *HS1* and *HS2*. The other, a 248 bp fragment (R6.22: 3R: 16748578–16748825), contains only the *HS1* region. These fragments were inserted in both the forward and reverse orientations. We also tested a 505 bp fragment spanning the *HS2* (R6.22: 3R: 16748826–16749330) in both the forward and reverse orientations (**Figure 4—figure supplement 2**). While *HS2* lacks blocking activity (**Figure 4—figure supplement 1** and **Figure 4—figure supplement 2**), **Figure 4B** shows that both of the *HS1* fragments block crosstalk between *iab-6* and *iab-7* and rescue the GOF transformation of A6 into A7. The rescuing activity is orientation independent. In all four

cases, however, the morphology of the A6 segment resembles that normally observed in A5, not A6. Instead of the banana-shaped A6 sternite observed in WT, the A6 sternite in *HS1* replacements has a quadrilateral shape and is covered in bristles just like the A5 sternite (*Figure 4*). A similar LOF A6 toward A5 transformation is evident in A6 tergite, which is covered in trichome hairs like A5 tergite (*Figure 4—figure supplement 1*). This LOF phenotype is observed when the *Fab-7* replacements have insulating activity but do not support bypass (*Hogga et al., 2001*; *Kyrchanova et al., 2017*). Note, however, that *HS1* replacements cannot block *iab-5* domain from activating *Abd-B*, and A5 segment identity is not changed. This is likely due to the fact that *Fab-6* boundary can 'bypass' this intervening insulator and target *iab-5* enhancers to *Abd-B* promoter (for detailed review, see *Kyrchanova et al., 2015*). Consistent with the adult phenotypes, *Abd-B* expression in the embryonic CNS is suppressed in A6 by the *HS1* fragments (*Figure 4C*).

dCTCF recognition sequences were found to be required for the blocking activity of other CTCF-associated BX-C boundaries (*Kyrchanova et al., 2017*; *Kyrchanova et al., 2016*, p. 8). This is also true for the dCTCF site in *HS1*. *Figure 4* shows that a mutation in the *HS1* dCTCF binding site completely disrupts insulating activity. Like the starting $F7^{attP50}$ platform, the adult $HS1^{358mCTCF}$ males lack the A6 segment, indicating that PS11/A6 is fully transformed into a copy of PS12/A7 (*Figure 4—figure supplement 1*). Consistent with a loss of blocking activity, the pattern of *Abd-B* expression in $HS1^{358mCTCF}$ embryos is similar to that of $F7^{attP50}$ (*Figure 4C*).

## Read-through transcription from *HS2* abrogates *HS1* insulator function

The finding that fragments containing only *HS1* have insulating activity, while the full-length *Fub-1* fragment does not, suggests that when *HS2* is present it inactivates the *HS1* boundary. One likely mechanism is transcriptional read-through. Studies on the induction of the chicken lysozyme gene by bacterial lipopolysaccharides (LPS) by Lefevre et al. showed that LPS treatment activated the transcription of an lncRNA through a CTCF boundary (*Lefevre et al., 2008*). Read-through transcription resulted in the inactivation of the boundary enabling distal enhancers to bypass the boundary and turn on the lysozyme gene.

To test this hypothesis, we inserted a 229 bp SV40 transcription terminator in between *HS2* and *HS1* to give the $HS2^{505R}$-SV40-$HS1^{248R}$ replacement (*Figure 5A*, *Supplementary file 1a*). Unlike the starting $HS2^{505R}$-$HS1^{248R}$ line, an A6-like segment is present in $HS2^{505R}$-SV40-$HS1^{248R}$ males (*Figure 5B* and *Figure 4—figure supplement 1*). However, blocking is not fully restored to the level of $HS1^{248R}$ alone. In $HS2^{505R}$-SV40-$HS1^{248R}$ males, the A6 tergite is slightly reduced in size while the sternite is absent. In the embryonic CNS, *Abd-B* expression in PS11 is clearly elevated, but less so than that observed in either $HS2^{505R}$-$HS1^{248R}$ or $F7^{attP50}$ (*Figure 5C* and *Figure 5—figure supplement 1*). To confirm that the partial reactivation of blocking activity is due to a reduction in transcriptional read-through, we generated a similar construct, $HS2^{505R}$-mSV40-$HS1^{248R}$ (*Supplementary file 1a*), in which the polyadenylation sequences in the SV40 fragment were mutated. In contrast to $HS2^{505R}$-SV40-$HS1^{248R}$, the insulating activity of $HS2^{505R}$-mSV40-$HS1^{248R}$ is substantially compromised and only a residual A6 tergite is observed (*Figure 5B* and *Figure 4—figure supplement 1*).

In these experiments, the SV40 termination element will not only suppress transcription through *HS1*, but also into the neighboring *iab-7* regulatory domain. To rule out the possibility that the SV40 element 'rescues' the blocking activity of *HS1* by reducing transcription into the *iab-7* regulatory domain, we placed it downstream of *HS1* instead between *HS2* and *HS1*. *Figures 5B and S4* show that just like $HS2^{505R}$-$HS1^{248R}$, $HS2^{505R}$-$HS1^{248R}$-SV40 has no boundary function: A6 is absent in males carrying the replacement, indicating that the cells in PS11/A6 have assumed a PS12/A7 identity. The *Abd-B* expression pattern in the embryonic CNS also closely matches that of $HS2^{505R}$-$HS1^{248R}$ line (*Figure 5C*). Similar results were obtained when the same construct ($SV40$-$HS1^{249}$-$HS2^{505}$) was inserted in the forward orientation so that the *HS2* element is 'pointing' toward the *iab-6* domain (*Figure 5* and *Figure 4—figure supplement 1*).

To provide further evidence that transcriptional read-through from *HS2* is responsible for disrupting the blocking activity of *HS1*, we generated a replacement, $HS2^{505}$-$HS1^{248R}$, in which the 5'→3' orientation of *HS2* was inverted with respect to *HS1*. If transcription from *HS2* is unidirectional, then the blocking activity of *HS1* should be unaffected when *HS2* is 'pointing' away from *HS1*. Consistent with this prediction, *HS1* retains blocking activity in $HS2^{505}$-$HS1^{248R}$. *Figure 5C* shows that the pattern of *Abd-B* expression in the embryonic CNS in both PS10 and PS11 is reduced by $HS2^{505}$-$HS1^{248R}$, while

the cuticle phenotype in adult males resembles $HS1^{248}$ (**Figure 5B**). Taken together, these results show that read-through transcription from *HS2* is responsible for the inactivation of *HS1* boundary function.

## Transcription from *HS2* promoter cannot overcome five dCTCF sites

Transcriptional read-through by RNA Polymerase II (Pol II) could disrupt *HS1* boundary activity by temporarily displacing dCTCF as well as other boundary associated factors. If this is the case, it seemed possible that the presence of additional dCTCF sites in *HS1* would counteract the effects of transcription from *HS2* either by helping to maintain boundary function as Pol II read-throughs or by inhibiting Pol II elongation. To test this possibility, we inserted four additional dCTCF sites in an *HS1* Sph I restriction site located close to the endogenous dCTCF site. **Figure 5B** shows that the addition of four dCTCF sites ($HS2^{505R}$-$HS1^{248R}$-$CTCF^{x4}$) is sufficient to rescue the blocking activity of *HS1*. As observed for *HS1* alone, $HS2^{505R}$-$HS1^{248R}$-$CTCF^{x4}$ males have a tergite and sternite whose morphology resembles that seen in A5. Likewise, the pattern of *Abd-B* expression in the CNS in $HS2^{505R}$-$HS1^{248R}$-$CTCF^{x4}$ is similar to that observed for $HS^{248}$ (**Figure 5C**).

## Segmental regulation of the *HS1* boundary by *HS2*-dependent transcriptional read-through

Taken together, the findings in the previous sections suggest that *Fub-1* likely functions as a boundary in its endogenous context; however, its boundary activity is segmentally regulated by transcriptional read-through from *HS2*. This model makes two predictions. First, the activity of the *HS2* promoter should be controlled by the *bxd/pbx* regulatory domain. Second, when transcriptional read-through is prevented, *Fub-1* blocking activity would be expected to interfere with *Ubx* regulation by the *bxd/pbx* regulatory domain. We have tested these predictions.

## *HS2* promoter activity is controlled by the *bxd/pbx* regulatory domain

In order to test the first prediction, we used smFISH probes spanning a 1983 bp region in the *bxd* domain located just proximal to *Fub-1* (**Figure 6A**, **Supplementary file 1b**). As noted above, Pease et al. found that there are two distinct 'sense' lncRNAs, *bxd* and $Fub-1^{HS2}$, expressed under the control of the *bxd/pbx* regulatory domain that could potentially read-through *HS1* and be complementary to sequences in the region between *HS1* and the *Ubx* promoter. For this reason, transcripts originating from the *HS2* promoter and extending through *HS1* toward the *Ubx* promoter cannot be unambiguously identified as being derived from *HS2* in *WT* using probes proximal to (downstream of) *Fub-1*.

As shown in **Figure 6—figure supplement 1**, we detected a broad band of sense transcripts in the posterior region of both *WT* and $F1^{attP}$ embryos as early as nuclear cycle 13. In *WT*, the smFISH probe is expected to hybridize to both $Fub-1^{HS2}$ and *bxd* lncRNAs. In contrast, since the $Fub-1^{HS2}$ promoter is deleted in $F1^{attP}$ embryos, only *bxd* lncRNAs should be detected. As would be expected if both *bxd* and $Fub-1^{HS2}$ lncRNAs are expressed at this stage in *WT* embryos, the intensity of the smFISH signal is higher in *WT* compared to $F1^{attP}$. During gastrulation, the broad band seen at nuclear cycle 13 resolves into a series of stripes of near equal width and intensity extending toward the posterior of the embryo (**Figure 6—figure supplement 2**). At this stage, the difference in the smFISH signal between *WT* and $F1^{attP}$ is even more striking than in pre-cellular blastoderm embryos. At around stage 11 of embryogenesis, the transcription of *bxd* lncRNAs seems to be shut off as we no longer detect lncRNAs complementary to the probe in $F1^{attP}$ embryos (**Figure 6—figure supplement 3**). In contrast, in *WT*, lncRNAs complementary to the probe are readily detected and are expressed in a pattern that resembles that of *Ubx* RNA, except that there is no staining in PS5/T3 (**Figure 6—figure supplement 3**). Thus, at this stage only the $Fub-1^{HS2}$ promoter seems to be active. Like the *DsRed* and *GFP* reporters described above (**Figure 3**), $Fub-1^{HS2}$ transcripts are detected in PS6/A1 through PS12/A7 (**Figure 6—figure supplement 3**). As shown in **Figure 6B** in stage 14 embryos, $Fub-1^{HS2}$ transcripts are detected in the epidermis in PS6–PS12, though the highest level is in PS6. This pattern of expression indicates that the *HS2* promoter is regulated by the *bxd/pbx* domain, while it is insulated from regulatory elements in the *abx/bx* domain.

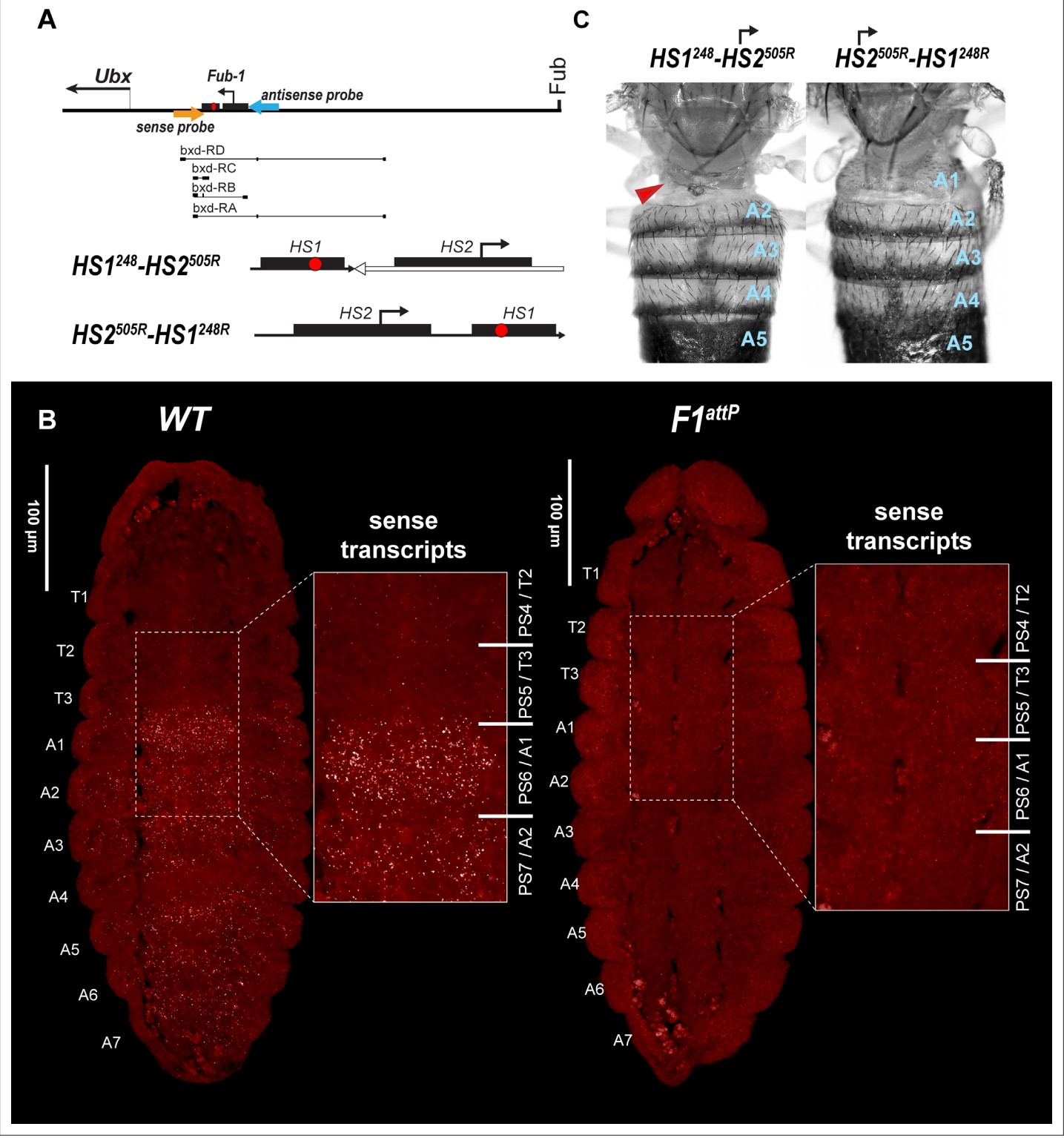

**Figure 6.** *HS2* drives transcription in tissue-specific manner. (**A**) Schematic presentation of *bxd* domain with *HS1²⁴⁸-HS2⁵⁰⁵ᴿ* and *HS2⁵⁰⁵ᴿ-HS1²⁴⁸ᴿ* replacements. The smFISH strand-specific probes are shown as orange and light blue arrows (***Supplementary file 1b and c***). All other designations are the same as described in ***Figure 1***. Characterized *bxd* lncRNA transcripts are presented under coordinate line. (**B**) smFISH of stage 14 embryos of the indicated genotype with a probe that targets sense strand just proximal to *Fub-1*. In *WT* embryos *Fub-1ᴴˢ²* sense (distal to proximal) transcripts were detected from PS6/A1 through PS12/A7. By contrast, in *Fub-1* deletion (*F1ᵃᵗᵗᴾ*) *Fub-1ᴴˢ²* transcripts were not detectable. (**C**) Morphology of the abdominal

*Figure 6 continued on next page*

*Figure 6 continued*

segments of *HS1²⁴⁸-HS2⁵⁰⁵ᴿ* and *HS2⁵⁰⁵ᴿ-HS1²⁴⁸ᴿ* flies. The red arrow shows the signs of the loss-of-function (LOF) phenotype: complete reduction of A1 segment and the appearance of postnatal tissue in its place.

The online version of this article includes the following figure supplement(s) for figure 6:

**Figure supplement 1.** Expression of *bxd* and *F1ᴴˢ²* lncRNAs in NC13 *WT* and *F1ᵃᵗᵗᴾ* embryos.

**Figure supplement 2.** Expression of *bxd* and *F1ᴴˢ²* lncRNAs in stage 7/8 *WT* and *F1ᵃᵗᵗᴾ* embryos.

**Figure supplement 3.** Expression of *bxd* and *F1ᴴˢ²* lncRNAs in stage 11 WT and *F1ᵃᵗᵗᴾ*embryos.

**Figure supplement 4.** Halters morphology in WT and *HS1²⁴⁸-HS2⁵⁰⁵ᴿ* flies.

**Figure supplement 5.** Homeotic transformation of larval segments in *HS1²⁴⁸-HS2⁵⁰⁵ᴿ*.

**Figure supplement 6.** *Ubx* expression in halter discs.

## Transcriptional read-through is required to abrogate *HS1* blocking activity in PS6/A1

The second prediction is that transcriptional read-through from *HS2* into *HS1* is required to relieve the blocking activity of *HS1*, enabling regulatory interactions between the *bxd/pbx* domain and *Ubx* in PS6/A1.

To test this prediction, we generated two different *Fub-1* replacements in which the 5′→3′ orientation of the *HS2* element was reversed so that it would generate 'antisense' transcripts. In the first, we inserted a fragment, *HS1²⁴⁸-HS2⁵⁰⁵ᴿ*, into *F1ᵃᵗᵗᴾ* that contains both *Fub-1* hypersensitive sites (*Figure 6A*). However, the 5′→3′ orientation of *HS2* in this fragment was reversed so that it is pointing away from *HS1*. In this case, transcripts expressed from the *HS2* promoter would extend into the *bxd/pbx* domain, instead of transcribing through *HS1*. In the second, we inserted a *Fub-1* fragment in *F1ᵃᵗᵗᴾ* containing both hypersensitive sites, *HS2⁵⁰⁵ᴿ-HS1²⁴⁸ᴿ*, but in the reverse orientation so that transcription from *HS2* would read-through *HS1* toward the *bxd/pbx* regulatory domain (*Figure 7A*). As shown in *Figure 6C*, the morphology of the A1 segment in *HS2⁵⁰⁵ᴿ-HS1²⁴⁸ᴿ* replacement flies is similar to that in *WT*. In this replacement transcripts originating in *HS2* read through *HS1* into the *bxd/pbx* regulatory domain. A different result is obtained for the replacement, *HS1²⁴⁸-HS2⁵⁰⁵ᴿ*, in which the *HS2* promoter is directed away from *HS1*. *HS1²⁴⁸-HS2⁵⁰⁵ᴿ* flies show evidence of an LOF transformation (*Figure 6C*). This replacement transforms the first abdominal segment into the third thoracic: the first abdominal tergite is reduced or absent; in addition, the posterior third thoracic segment is partly transformed to a posterior second thoracic segment: some flies have enlarged and/or pointing downward halters (*Figure 6C* and *Figure 6—figure supplement 4*). These phenotypes are known as *bithoraxoid* (*bxd*) and *postbithorax* (*pbx*), respectively (*Lewis, 1963*). *HS1²⁴⁸-HS2⁵⁰⁵ᴿ* homozygotes also display sterility. Homozygous flies showed very low viability, and we were able to generate homozygote larvae and flies only when growing heterozygotes on rich medium (*Backhaus et al., 1984*).

*HS1²⁴⁸-HS2⁵⁰⁵ᴿ* mutants also have an LOF phenotype at the larval stage. As shown in Figure S11, wild-type larvae have 11 denticle belts marking the anterior half of each of the thoracic and abdominal segments; these are visible as black dots. In *WT*, T3 has 2–3 rows of small denticles, whereas in A1 there are 3 anterior rows of large and 2 rows of small denticles. In the *HS1²⁴⁸-HS2⁵⁰⁵ᴿ* replacement, the A1 band of denticles is narrower and irregular, indicative of an LOF transformation of A1 toward T3 (*Figure 6—figure supplement 5*). We also observe a reduction in the expression of *Ubx* in the posterior compartment of the haltere discs compared to *WT* (*Figure 6—figure supplement 6*).

Since the *HS2* promoter is inserted in the reverse orientation in both *HS1²⁴⁸-HS2⁵⁰⁵ᴿ* and *HS2⁵⁰⁵ᴿ-HS1²⁴⁸ᴿ*, it is expected to generate antisense transcript in response to regulatory elements in *bxd/pbx*. To test this prediction, we generated antisense smFISH probes spanning a 1463 bp *bxd* sequence located just distal to *Fub-1* (*Figure 6A*, *Supplementary file 1c*). *Figure 7* shows that as expected antisense transcripts complementary to the smFISH probes are not detected in *WT* embryos. In contrast, segmentally restricted antisense transcripts are expressed in both of the *Fub-1* replacements. Like *Fub-1ᴴˢ²* sense transcripts in *WT* (*Figure 6*), the antisense transcripts are detected in PS6/A1 through PS12/A7 in *HS1²⁴⁸-HS2⁵⁰⁵ᴿ* and *HS2⁵⁰⁵ᴿ-HS1²⁴⁸ᴿ*. The transcripts levels in *HS1²⁴⁸-HS2⁵⁰⁵ᴿ* and *HS2⁵⁰⁵ᴿ-HS1²⁴⁸ᴿ* embryos are not, however, equivalent. The signal is noticeably greater in *HS1²⁴⁸-HS2⁵⁰⁵ᴿ* than it is in *HS2⁵⁰⁵ᴿ-HS1²⁴⁸ᴿ*. This could reflect the fact that the smFISH probes are complementary to sequences immediately adjacent to *HS2* in the *HS1²⁴⁸-HS2⁵⁰⁵ᴿ* replacement, whereas the polymerase

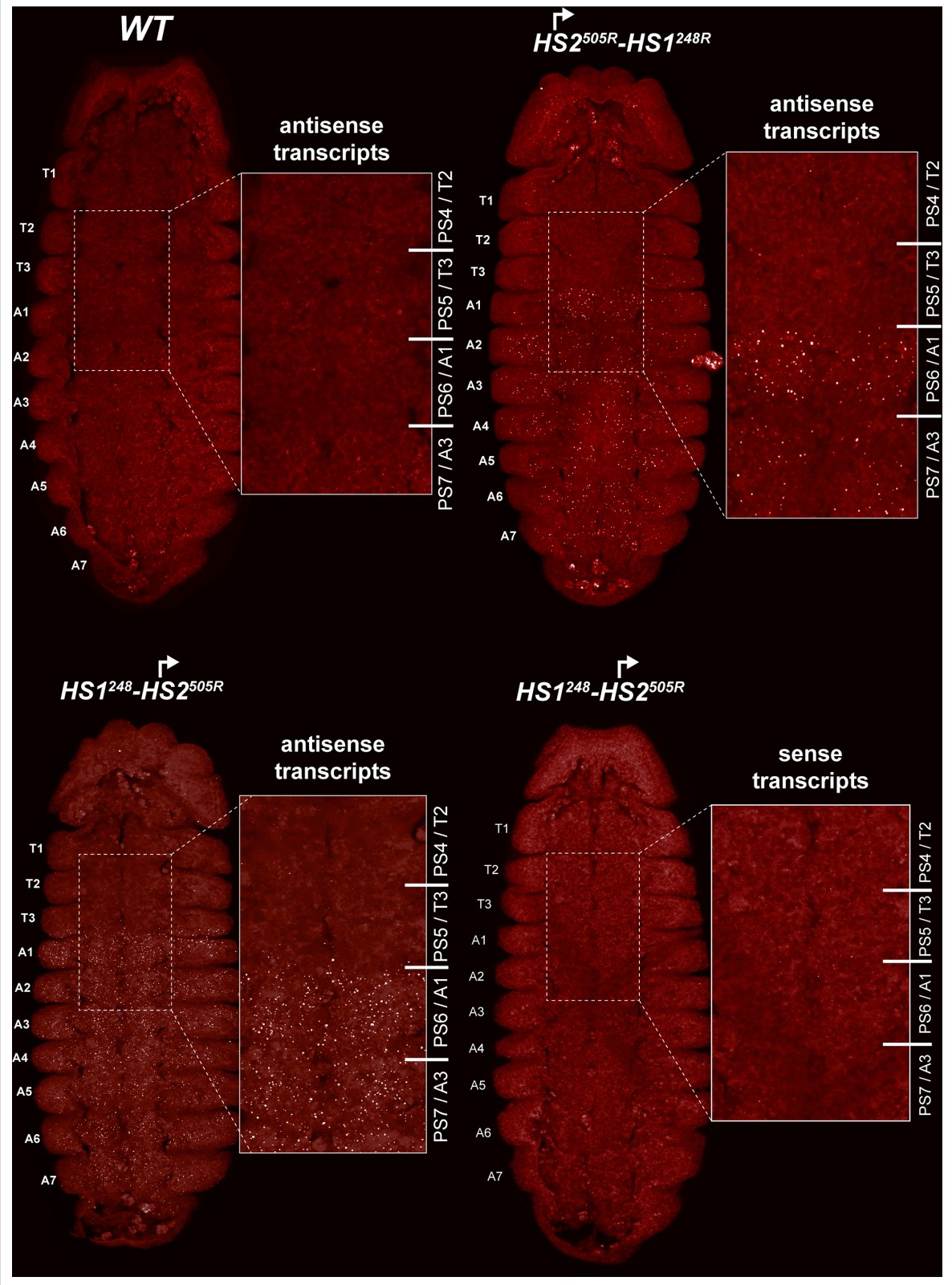

**Figure 7.** *HS2* drives transcription in a tissue-specific manner. smFISH of stage 14 embryos of indicated genotype with a probe that targets antisense strand just distal to *Fub-1*. In *WT* embryos, no antisense (proximal to distal) transcripts were detected. By contrast, in *HS1^248^-HS2^505R^* and *HS2^505R^-HS1^248R^* antisense transcripts are detected from PS6/A1 through PS12/A7. Note that the intensity of smFISH signal is higher in *HS1^248^-HS2^505R^* than in *HS2^505R^-HS1^248R^*. In *HS1^248^-HS2^505R^* embryos, *Fub-1^HS2^* sense (distal to proximal) transcripts were not detected, indicating that *HS2* drives transcription only in one direction.

must traverse *HS1* in the *HS2505R*-*HS1248R* replacement. Alternatively, since *HS1* is located between the *HS2* promoter and the *bxd/pbx* regulatory domain, it may attenuate regulatory interactions, and at least the initial activation in PS6/A1 may depend upon the enhancers thought to be located between the *Ubx* promoter and *Fub-1*.

In order to test whether *HS2* drives transcription in both directions, we assayed *HS1248*-*HS2505R* with the probe targeting sense strand. **Figure 7** shows that, unlike *WT* (**Figure 6**), sense transcription is not detectable when *HS2* is inverted. This result indicates that the lncRNA promoter is situated in *HS2* and drives transcription only in one direction.

## Discussion

While a very large percentage of the Pol II transcripts in multicellular eukaryotes are lncRNAs, what roles they play in the expression of coding mRNAs and in development is largely unknown. Here we have investigated the functional properties of an lncRNA gene, *Fub-1HS2*, that is located in the *bxd/pbx* regulatory domain of *Drosophila* BX-C. As has been observed for many other lncRNA genes, the RNA sequences encoded by *Fub-1HS2* are not well conserved and it is the functional properties of the two *Fub-1HS2* regulatory elements, *HS1* and *HS2*, that are important. We show here that *HS1* is a dCTCF-dependent boundary element, while *HS2* is a developmentally regulated promoter that directs transcription through *HS1* inactivating its boundary function in parasegments that require *Ubx* function.

The arrangement of the three homeotic genes and the nine parasegment-specific regulatory domains in the *Drosophila* BX-C complicates the coordination between the 3D organization of the complex and the requirements for regulatory interactions. In order to specify segment identity, the nine regulatory domains are segregated into topologically independent loops by boundary elements. This physical organization helps to block crosstalk between initiation elements in adjacent domains in the early embryo, while later in development it helps restrict the spread of PcG silencing from inactive to active domains (**Kyrchanova et al., 2015**; **Maeda and Karch, 2015**). However, in order to direct the proper expression of their target homeotic gene, all but three of the regulatory domains must be able to bypass one or more intervening boundary elements. In the case of the *Ubx* gene, the *bxd/pbx* regulatory domain is separated from its target promoter by the *Fub-1* boundary and by a second boundary element located close to the *Ubx* promoter.

The *Fub-1* boundary is subdivided into two elements, *HS1* and *HS2. HS1,* like other boundaries in BX-C, has a dCTCF site, is bound by dCTCF in vivo, and is able to function as a generic insulator. While *HS2* may contribute to the boundary function of *Fub-1*, it does not have insulating activity on its own. Instead, it has promoter activity and can function as an enhancer trap. In its normal context, the *HS2* promoter is controlled by the *bxd/pbx* regulatory domain. The *bxd/pbx* regulatory domain is set in the *off* state in parasegments (segments) anterior to PS6/A1 at the blastoderm stage and is repressed during the remainder of development by PcG-dependent silencing. In PS6/A1 (and more posterior segments), the *bxd/pbx* regulatory domain is set in the *on* state, activating the stage and tissue-specific enhancers in this domain. These stage- and tissue-specific enhancers activate the *HS2* promoter in PS6/A1 and more posterior parasegments and transcription from this promoter through *HS1* disrupts *Fub-1* boundary function (**Figure 8**). This enables stage- and tissue-specific enhancers in *bxd/pbx* to activate *Ubx* expression in PS6/A1 and more posterior parasegments.

Several lines of evidence are consistent with this mechanism. When *HS1* is used to replace the *Fab-7* boundary, it functions like other generic fly boundaries and blocks crosstalk between the *iab-6* and *iab-7* initiators, but does not support boundary bypass. This blocking activity is also orientation independent. In contrast, neither *HS2* alone nor the intact *Fub-1* (*HS1*+*HS2*) element are able to block crosstalk between *iab-6* and *iab-7* and both replacements exhibit the same GOF transformations as the starting *F7attP* replacement platform. In the *HS1*+*HS2 Fab-7* replacement experiments, the defects in *Fub-1* blocking activity arise from transcriptional read-through from the *HS2* promoter. This conclusion is supported by two lines of evidence. First, introducing the SV40 transcription termination polyadenylation element in between *HS2* and *HS1* partially rescues the blocking defect. Rescuing activity is not due to the increased distance between *HS2* and *HS1* as a SV40 element with mutations in the polyadenylation signal significantly compromising its rescuing activity. In addition, the SV40 termination element must be located between *HS1* and *HS2* as rescue is not observed when it is

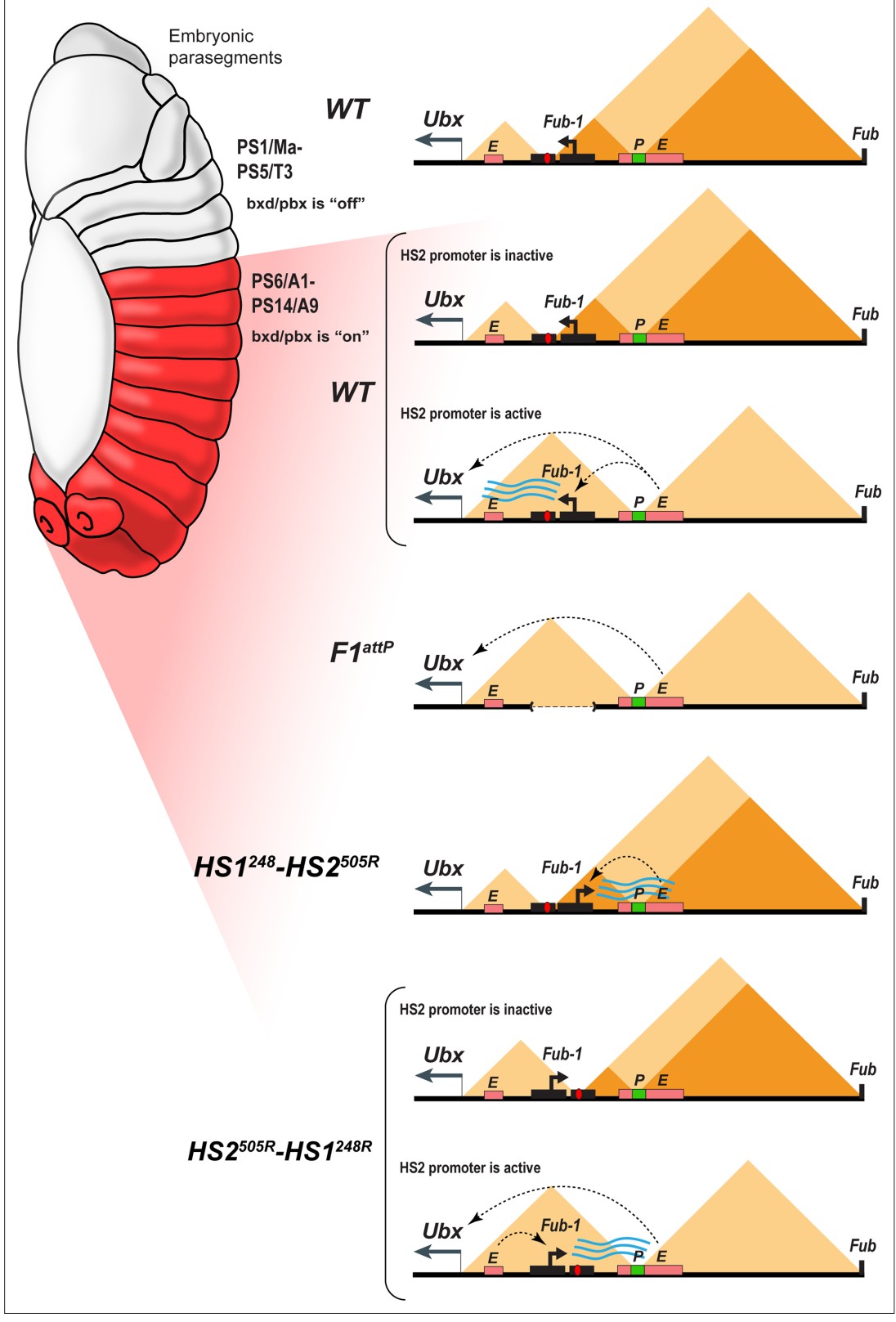

**Figure 8.** Model showing changes in TAD organization during *Fub-1^HS2* transcription. Top: In *WT*, the 3D organization of the *bxd/pbx* regulatory domain transitions from one state to another when it activates *Ubx* expression (see also ***Mateo et al., 2019***). In PS5 (T3) and more anterior parasegments, the *bxd/pbx* domain is encompassed in a large TAD that extends from *Fub-1* to *Fub* (large triangle). Within this large TAD, there are two HDIC domains. One extends from *Fub-1* to the *bxd* PRE, while the other extends from the *bxd* PRE to *Fub*. In PS6 and more posterior parasegments, the 3D

*Figure 8 continued on next page*

*Figure 8 continued*

organization of the domain is dynamic and depends upon whether the *Fub-1 HS2* promoter is active or not. When the promoter is inactive (in between bursts), the *Fub-1* boundary is expected to be functional and that TAD/HDIC organization would resemble that in more anterior parasegments. When the enhancers in *bxd/pbx* activate *HS2*, transcriptional read-through disrupts *Fub-1* boundary activity generating a new TAD organization that links enhancers in *bxd/pbx* to the *Ubx* promoter. *F1^{attP}*: deletion of the *Fub-1* boundary eliminates the large *Fub-1←→Fub* TAD in all parasegments. In this case, an element upstream of the *Ubx* promoter functions to insulate *abx/bx* from *bxd/pbx* in PS5/T3. *HS1^{248}-HS2^{540R}*: In this replacement, *HS2* is inverted so that the *HS2* promoter is pointed away from *HS1* toward the *bxd/pbx* regulatory domain. Though *bxd/pbx* activates the promoter in PS6/A1 and more posterior parasegments, *Fub-1* boundary activity is not disrupted. *HS2^{540R}-HS1^{248R}*: In this replacement, *Fub-1* is inverted. Though *HS2* promoter activity in PS6 (A1) and more posterior parasegments is reduced in this configuration compared to WT, it is sufficient to disrupt *Fub-1* boundary activity and enable *bxd/pbx* to regulate *Ubx* expression.

placed downstream of *HS1*. Second, when *HS2* is inverted so that transcription proceeds away from *HS1*, boundary function is also restored.

A similar mechanism appears to regulate *Fub-1* boundary activity in its endogenous context; however, in this case boundary activity is segmentally regulated. In PS5 and more anterior parasegments, the *bxd/pbx* regulatory domain is maintained in an '*off*' state by a PcG-based mechanism. As a consequence, the *Fub-1 HS2* promoter is inactive (*Figure 8*). In PS6 (A1) (and more posterior parasegments), the *bxd/pbx* domain is in the '*on*' state and it activates transcription from the *Fub-1 HS2* promoter. When *HS2* is oriented so that transcription proceeds through *HS1*, boundary activity is abrogated in PS6/A1 and more posterior parasegments/segments (*Figure 8*). This enables enhancers in the *bxd/pbx* regulatory domain to activate *Ubx* transcription. Consistent with the parasegment-specific differences in the 3D topology imaging studies of fly embryos, Mateo et al. showed that the *Ubx* region of BX-C undergoes a reorganization in PS5/T3 and PS6/A1 (*Mateo et al., 2019*). This rearrangement would be blocked when *HS2* is oriented away from *HS1*. In this case, boundary activity is retained and the stage and tissue enhancers in the *bxd/pbx* regulatory domain are unable to activate *Ubx* expression (*Figure 8*).

While it is clear that the *Fub-1* boundary function is subject to negative and positive regulation by elements in the *bxd/pbx* domain, and that this is important for the proper specification of PS6/A1, there are a number of outstanding questions. One is whether transcription from the *HS2* promoter is necessary to inactivate *HS1* earlier in development. As described above, *bxd* lncRNA transcripts emanating from a promoter in *pbx* extend through both *HS2* and *HS1*. If *HS1* is sensitive to read-through from *HS2*, one would think that it would also be disrupted by transcriptional read-through of the *bxd* lncRNA. However, since Pease et al. found that transcription of the *bxd* lncRNA is not required for normal development, it would appear that this lncRNA is dispensable, at least when *Fub-1 HS2* is present. While we do not know the pattern of *Fub-1* or *bxd* lncRNAs expression after embryogenesis, the fact that reversing *HS2* results in morphological defects in adults would argue that *Fub-1* is likely expressed during the larval and pupal stages.

Another question is raised by the Micro-C contact pattern in *WT* and the *Fub-1* deletion. In *WT*, *Fub-1* and *Fub* define a TAD that encompasses (most of) *bxd/pbx*. *Fub-1* also corresponds to a boundary for two smaller HDIC domains, one with a second endpoint close to the *Ubx* promoter and another with a second endpoint close to the *bxd* PRE. In the *Fub-1* deletion, the proximal endpoint of the subTAD formed by *Fub-1* and *bxd* PRE shifts to the *Ubx* promoter region, while the two HDIC domains collapse into one. If the function of *HS2* is to inactivate *HS1* in the posterior 2/3rds of the embryo, one would expect that this would disrupt the *Fub-1 ←→ Fub* TAD, much like the *Fub-1* deletion. However, we observe a *Fub-1 ←→ Fub* TAD. Since the *HS2* promoter should not be active in cells anterior to PS6/A1, it is possible that these cells are responsible for generating the *Fub-1 ←→ Fub* TAD seen in the Micro-C experiments. Alternatively (or in addition), the disruption in *HS1* function might only occur during a burst of transcription (*Figure 8*). Recent studies have shown that transcription of most genes is not continuous but rather occurs in discrete bursts (*Rodriguez and Larson, 2020*; *Tunnacliffe and Chubb, 2020*). The frequency, duration, and amplitude of the burst can vary substantially. In some experimental systems, bursts can last several minutes or longer while the intervals between bursts may be only 10s of seconds (*Bothma et al., 2014*; *Fukaya et al., 2016*). For other genes, the intervals between bursts can be 30 min or more and the bursts may last only a minute or two (*Alexander et al., 2019*). If disruption of *HS1* blocking activity is directly coupled to the act of transcription (as appears to be the case), then there may only be small windows of time

in which the activity of the *Fub-1* boundary is compromised and *bxd/pbx* enhancers can activate the *Ubx* promoter. In this case, the *Fub-1* insulator would flip back and forth from an inactive and active state (*Figure 8*, see *WT* and *HS2^540R^-HS1^248R^*). Potentially consistent with a model along these lines, in stage 14 embryos, only a subset of cells in PS6 and more posterior parasegments have transcripts complementary to *Fub-1^HS2^* (*Figure 6*).

A related issue would be the mechanism activating *HS2* transcription when the *Fub-1* boundary is reversed (the *HS1^248^-HS2^505R^* replacement) so that *HS1* is interposed between *HS2* and the *bxd/pbx* domain. One would have expected that *HS1* would block the *bxd/pbx* enhancers from activating *HS2* transcription. In this case, boundary function would be retained, resulting in an LOF transformation of PS6/A1 into PS5/T3 because it would prevent *bxd/pbx* from regulating *Ubx*. One likely possibility is that the putative enhancers in the region between the *Ubx* promoter and *Fub-1* (see *2218S* enhancer in *Figure 1—figure supplement 1*) are able to activate a sufficient level of *HS2*-dependent transcription to disrupt *HS1* boundary function. This is supported by the pattern of expression of reporters inserted to either side of *Fub-1* (*Figure 3*). An alternative/additional possibility is that the blocking activity of *HS1* is not sufficient to completely suppress regulatory interactions between *bxd/pbx* and *HS2*. Potentially consistent with this idea is the finding that the orientation of the *Fub-1* element in the *Fab-7* replacements does not seem to be important. When *HS2* is adjacent to *iab-6*, it inactivates the blocking activity of *HS1*. This would be the expected result since reporters in BX-C are activated when the regulatory domain in which they are inserted is turned *on* in the early embryo. In this case, *iab-6* would drive *HS2* expression in PS11(A6) cells and this would inactivate the blocking activity of *HS1* in cell in this parasegment, giving a GOF transformation of PS11/A6. The unexpected result is that a GOF transformation of PS11/A6 into PS12/A7 is also observed when the *Fub-1* replacement is inserted so that *HS2* is next to *iab-7*. Since *iab-7* is 'off' in PS11, this would imply that *HS2* must be activated by the *iab-6* regulatory domain. A similar scenario could be at play when *Fub-1* is inserted in the opposite orientation in its endogenous location: *HS1* blocks but does not completely eliminate interactions between the *bxd/pbx* enhancers and the *HS2* promoter. This is probably not unexpected. While interposed boundaries significantly reduce enhancer:promoter contacts, they are not completely eliminated (*Fukaya et al., 2016*). In addition, once activated, transcription from *HS2* through *HS1* should weaken *HS1* boundary activity and this in turn would facilitate further transcription.

Perhaps the most puzzling question is what function does *Fub-1* play in BX-C regulation? Since there is no obvious GOF phenotype when *Fub-1* is deleted, it differs from other boundaries in BX-C that have been analyzed in that would appear to be dispensable. On the other hand, when present, it must be inactivated in PS6/A1 (and more posterior parasegments) so that the *bxd/pbx* domain can regulate *Ubx* expression. Interestingly, like *cis*-acting elements for many other lncRNAs, both of the *Fub-1* boundary subelements are conserved in the *Sophorphora* subgenus (*Figure 1—figure supplement 3*). It is also conserved in *D. virilis,* which is a member of the *Drosophila* subgenus. *D. virilis* differs from *melanogaster* in that the *Ubx* gene and its two regulatory domains, *abx/bx* and *bxd/pbx*, are separated from the *abd-A* and *Abd-B* genes. In spite of this rearrangement, the *Fub-1* sequences are conserved. The fact that *Fub-1* is present in distantly related species would argue that *Fub-1* has an important function under some type of selective condition (that we have yet to discover) that is commonly encountered by a wide range of *Drosophila* species.

## Materials and methods
### Fly strains
The following stocks were used in these experiments: y[1] M{GFP[E.3xP3=vas-Cas9.RFP-}ZH-2A w[1118] (BDSC_55821), w[1], ΦC31 (y⁺); TM6, Tb, Hm/Sb, w[1]; TM6, Tb, Hm/Sb, P{w[+mC]=ovo-FLP.R}M1A, w[*] (BDSC_8727), y[1] w[67c23] P{y[+mDint2]=Crey}1b; D[*]/TM3, Sb[1] (BDSC_851) was used for all transgenesis. y[1] M{GFP[E.3xP3]=vas-Cas9.RFP-}ZH-2A w[1118] (BDSC_55821) was used for CRISPR/Cas9 genome editing. y[1] w[67c23] P{y[+mDint2]=Crey}1b; D[*]/TM3, Sb[1] (BDSC_851) was used for Cre/loxP recombination to remove DsRed from CRISPR-generated DsRed insertions. w[1]; TM6, Tb, Hm/Sb, P{w[+mC]=ovoFLP.R}M1A, w[*] (BDSC_8727) was used for Flp/frt recombination to remove GFP from ΦC31-mediated insertions. *HS1^248^-HS2^505R^* stock is weak and was grown on rich medium described by *Backhaus et al., 1984*. All other stocks were kept on standard fly cornmeal-molasses-yeast media.

## Generation of *F1*[attP] by CRISPR-Cas9-induced homologous recombination

For generating double-stranded DNA donor template for homology-directed repair, we used pHD-DsRed vector (Addgene plasmid no. 51434). The final plasmid contains genetic elements in the following order: [bxd proximal arm]-[attP]-[loxP]-[3×P3-dsRed-SV40polyA]-[loxP]-[bxd distal arm] (*Figure 1—figure supplement 1*). Homology arms were PCR-amplified from the Bloomington Drosophila Stock Center line no. 55821 genomic DNA using the following primers: AGCTCTCG AGAGCGATGGAACCGTTTCTG and GTAGATCTCGGTTTACGATCGACTGGC for the proximal arm (1001 bp fragment), and TTGCGGCCGCATTTCTACGTTTATAAGCTGTTAATC and AAGCATGC CCACAATGGAGGCTGC for the distal arm (1002 bp fragment). Targets for Cas9 were selected using 'CRISPR optimal target finder'—the program from O'Connor-Giles Laboratory. The following gRNA sequences, AGTCGATCGTAAACCGGAG and TTATAGACGTAGAAATGTA, were inserted into the pU6-BbsI-chiRNA vector (Addgene plasmid no. 45946). A mixture of the donor vector (500 ng/ml) and two gRNA vectors (100 ng/ml each) was injected into the embryos of y[1] M{GFP[E.3xP3]=vas-Cas9. RFP-}ZH-2A w[1118] (no. 55821 from the Bloomington Drosophila Stock Center). Injectees were grown to adulthood and crossed with w[1]; TM6/Sb line. Flies positive for dsRed signal in eyes were selected into a new separate line. The successful integration of the recombination plasmid was verified by PCR and corresponded to the removal of 1168 bp within the *Fub-1* region (genome release R6.22: 3R:16,748,143.16,749,310).

## Generation the replacement lines

For the *F1*[attP] replacement, the recombination plasmid was designed de novo and contains several genetic elements in the following order: [loxP]-[pl]-[frt]-[3P3-GFP]-[frt]-[attb] (*Figure 1—figure supplement 1*). All elements were assembled within the pBluescript SK vector. loxP site is located before polylinker [pl] and in combination with the second site, which is located in the platform, used for excision of *DsRed* marker gene and plasmid body. Two frt sites are used to excise GFP marker. DNA fragments used for the replacement experiments were generated by PCR amplification and verified by sequencing (presented in the Supplementary Methods).

The strategy of the Fab-7 replacement lines creation is described in detail in *Wolle et al., 2015*.

## Adult and larval cuticle samples preparation

Three-day adult male flies were collected in 1.5 ml tubes and stored in 70% ethanol at least 1 d. Then ethanol was replaced with 10% KOH and flies were heated at 70°C for 1 hr. After heating, flies were washed with dH$_2$O two times and heated again in dH$_2$O for 1 hr. After that, the digested flies were washed three times with 70% ethanol and stored in 70% ethanol. The abdomen cuticles were cut off from the rest of the fly using a fine tweezer and a needle of an insulin syringe and put in a droplet of glycerol on a microscope slide. Then the abdomens were cut longitudinally on the mid-dorsal side through all of the tergites with the syringe. Then the cuticles were flattened with a coverslip. *HS1*[248]-*HS2*[505R] third instars were used for larval cuticle preparation. This stock is homozygous sterile and was maintained on TM6B balancer. The TM6B balancer carries the Tb[1] dominant mutation, which results in tubby larvae and pupae. Larvae without tubby phenotype were selected for cuticle preparation. Larval cuticle was prepared as described by *Ingham, 1984*. Photographs in the bright or dark field were taken with the Nikon SMZ18 stereomicroscope using Nikon DS-Ri2 digital camera, processed with Fiji bundle 2.0.0-rc-46.

## Embryo immunostaining

Embryo immunostaining was performed as previously described (*Deshpande et al., 1999*). Primary antibodies were mouse monoclonal anti-Abd-B at 1:40 dilution (1A2E9, generated by S. Celniker, deposited to the Developmental Studies Hybridoma Bank), mouse monoclonal anti-DsRed at 1:50 dilution (Santa Cruz Biotechnology), and rabbit polyclonal anti-GFP at 1:500 dilution (Thermo Fisher Scientific). Secondary antibodies were goat anti-mouse Alexa Fluor 546 (Thermo Fisher Scientific) and goat anti-rabbit Alexa Fluor 488 (Thermo Fisher Scientific). Stained embryos were mounted with VECTASHIELD antifade mounting medium (Vector Laboratories). Images were acquired with a Leica STELLARIS 5 confocal microscope and processed using ImageJ 1.53f51 (*Schindelin et al., 2012*).

## smFISH

smFISH was performed as previously described (*Little et al., 2013*). Quasar 670-conjugated Fub-1 sense and antisense probes were ordered from LGC Biosearch Technologies. Sequences are listed in *Supplementary file 1b and c*. Embryos were collected overnight and then dechorionated in Clorox, fixed in 4% paraformaldehyde, and devitellinized in methanol. The hybridization with the FISH probes was performed overnight at 37°C in hybridization buffer (10% dextran sulfate, 0.01% salmon sperm single-strand DNA, 1% vanadyl ribonucleoside, 0.2% bovine serum albumin, 4×SSC, 0.1% Tween-20, and 35% formamide). The resulting preparations were washed twice with wash buffer, 1 hr each, at 37°C and mounted in VECTASHIELD antifade mounting medium. Images were acquired with a Leica STELLARIS 5 confocal microscope with ×40 HC PL APO CS2 1.3 NA oil immersion objective. Image denoising was performed using NIS elements 'Advanced denoising' tool.

## Micro-C library construction

Embryos were collected on yeasted apple juice plates in population cages. Plates were laid for 4 hr, incubated for 12 hr at 25°C, then subjected to fixation. Embryos were collected in nylon mesh sieves, dechorionated for 2 min in 3% sodium hypochlorite, rinsed with deionized water, and transferred to glass vials containing 5 ml PBST (0.1% Triton-X in PBS), 7.5 ml N-heptane, and 1.5 ml fresh 16% formaldehyde. Crosslinking was carried out at room temperature for exactly 15 min on an orbital shaker at 250 rpm, followed by addition of 3.7 ml 2 M Tris-HCl pH7.5 and shaking for 5 min to quench the reaction. Embryos were washed twice with 15 ml PBST and subjected to secondary crosslinking. Secondary crosslinking was done in 10 ml of freshly prepared 3 mM final DSG and ESG in PBST for 45 min at room temperature with passive mixing. Reaction was quenched by addition of 3.7 ml of 2 M Tris-HCl pH7.5 for 5 min, then washed twice with PBST. Embryos were snap-frozen and stored at –80°C until library construction.

Micro-C libraries were prepared as previously described (*Batut et al., 2022*) with the following modification: we used 50 ul of 12–16 hr embryos, non-sorted for each biological replicate. 60U of MNase was used for each reaction to digest chromatin to 80% mononucleosome vs. 20% dinucleosome ratio. Libraries were barcoded, pooled, and subjected to paired-end sequencing on an Illumina Novaseq S1 100 nt Flowcell (read-length 50 bases per mate, 6-base index read).

## Micro-C data processing

Micro-C data for *Drosophila melanogaster* were aligned to a custom genome edited from the Berkeley Drosophila Genome Project (BDGP) Release 6 reference assembly (*dos Santos et al., 2015*) with BWA-MEM (*Li and Durbin, 2009*) using the parameters -S -P -5 -M. Briefly, the custom genome was designed to remove the visually obstructive ~5.6 kb DIVER_I-int repeat element between Fub-1 and Fub. The specific region deleted was the entire repeat element, chr3R: 16756620–16762262 from the dm6 reference FASTA file. The resultant BAM files were parsed, sorted, de-duplicated, filtered, and split with pairtools (https://github.com/open2c/pairtools; *Goloborodko et al., 2023*). We removed pairs where only half of the pair could be mapped or where the MAPQ score was less than three. The resultant files were indexed with pairix (https://github.com/4dn-dcic/pairix; *Lee et al., 2021*). The files from replicates were merged with pairtools before generating 100 bp contact matrices with Cooler (*Abdennur and Mirny, 2020*). Finally, balancing and mcool file generation was performed with Cooler's zoomify tool. The mcool files were visualized using HiGlass (*Kerpedjiev et al., 2018*).

## Acknowledgements

We are grateful to the Center for Precision Genome Editing and Genetic Technologies for Biomedicine of IGB RAS and the Core Facilities Center of IGB RAS for providing research equipment. We thank Tatyana Smirnova for technical assistance and advice on smFISH data acquisition and analysis. We thank Girish Deshpande, Tsutomu Aoki, Olga Kyrchanova, and Oleg Bylino for insightful discussions throughout the course of this work. This work was supported by R35 GM126975 to PS and R01 GM118147 to ML. Part of this work (cloning) was supported by the Russian Science Foundation, Project No. 20-14-00201 (to YS). Part of this work (transgenesis) was supported by the Russian Science Foundation (19-14-00103) and grant no. 075-15-2019-1661 from the Ministry of Science and Higher Education of the Russian Federation (immunohistochemistry).

## Additional information

### Competing interests

Xin Yang Bing: is affiliated with BlueRock Therapeutics. The author has no other competing interests to declare. The other authors declare that no competing interests exist.

### Funding

| Funder | Grant reference number | Author |
|---|---|---|
| National Institute of General Medical Sciences | R35 GM126975 | Paul Schedl |
| National Institute of General Medical Sciences | R01 GM118147 | Michael Levine |
| Russian Science Foundation | 20-14-00201 | Yulii V Shidlovskii |
| Russian Science Foundation | 19-14-00103 | Pavel Georgiev |
| Ministry of Science and Higher Education of the Russian Federation | 075-15-2019-1661 | Pavel Georgiev |

The funders had no role in study design, data collection and interpretation, or the decision to submit the work for publication.

### Author contributions

Airat Ibragimov, Conceptualization, Data curation, Formal analysis, Validation, Investigation, Visualization, Methodology, Writing - original draft, Writing - review and editing; Xin Yang Bing, Data curation, Validation, Investigation, Visualization; Yulii V Shidlovskii, Michael Levine, Resources, Supervision, Funding acquisition, Project administration; Pavel Georgiev, Conceptualization, Resources, Formal analysis, Supervision, Funding acquisition, Validation, Methodology, Project administration, Writing - review and editing; Paul Schedl, Conceptualization, Resources, Formal analysis, Supervision, Funding acquisition, Validation, Methodology, Writing - original draft, Project administration, Writing - review and editing

### Author ORCIDs

Airat Ibragimov  http://orcid.org/0000-0002-5973-9147
Yulii V Shidlovskii  https://orcid.org/0000-0002-3643-9889
Paul Schedl  http://orcid.org/0000-0001-5704-2349

### Decision letter and Author response

Decision letter https://doi.org/10.7554/eLife.84711.sa1
Author response https://doi.org/10.7554/eLife.84711.sa2

## Additional files

### Supplementary files

• Supplementary file 1. Sequences of DNA fragments used. (**a**) The sequence of chimeric DNA fragments used for *Fab-7* replacement experiments. (**b**) Sequences of *Fub-1 sense* smFISH probes covering 1983 bp region in *bxd* domain. (**c**) Sequences of *Fub-1 antisense* smFISH probes covering 1463 bp region in *bxd* domain.

• MDAR checklist

### Data availability

Sequencing data have been deposited in GEO under accession code GSE217005. The data can be accessed using the following secure token sxyxomkyjlgxvmj.

The following dataset was generated:

| Author(s) | Year | Dataset title | Dataset URL | Database and Identifier |
|---|---|---|---|---|
| Ibragimov A, Bing XY, Shidlovskii Y, Levine M, Georgiev P, Schedl P | 2022 | The insulating activity of the *Drosophila* BX-C chromatin boundary Fub-1 is parasegmentally regulated by lncRNA read-through | https://www.ncbi.nlm.nih.gov/geo/query/acc.cgi?acc=GSE217005 | NCBI Gene Expression Omnibus, GSE217005 |

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

# Appendix 1

**Appendix 1—key resources table**

| Reagent type (species) or resource | Designation | Source or reference | Identifiers | Additional information |
|---|---|---|---|---|
| Genetic reagent (*Drosophila melanogaster*) | Cas9 | Bloomington Drosophila Stock Center | BDSC: 55821 | y[1] M{GFP[E.3xP3]=vas-Cas9.RFP-}ZH-2A w[1118] |
| Genetic reagent (*D. melanogaster*) | ΦC31 | This lab | N/A | w[1], ΦC31 (y⁺); TM6, Tb, Hm/Sb |
| Genetic reagent (*D. melanogaster*) | | This lab | N/A | w[1]; TM6, Tb, Hm/Sb |
| Genetic reagent (*D. melanogaster*) | Fub-1[attP] | This paper | N/A | Available from Paul Schedl laboratory in Princeton university. Contact pschedl@princeton.edu |
| Genetic reagent (*D. melanogaster*) | Fab-7[attP50] | This lab | N/A | Available from Paul Schedl laboratory in Princeton university. Contact pschedl@princeton.edu |
| Genetic reagent (*D. melanogaster*) | FLP | Bloomington Drosophila Stock Center | BDSC: 8727 | P{w[+mC]=ovoFLP.R}M1A, w[*] |
| Genetic reagent (*D. melanogaster*) | Cre | Bloomington Drosophila Stock Center | BDSC: 851 | y[1] w[67c23] P{y[+mDint2]=Crey}1b; D[*]/TM3, Sb[1] |
| Antibody | Anti- Abd-B (mouse monoclonal) | Developmental Studies Hybridoma Bank | 1A2E9 | IF(1:50) |
| Antibody | Anti-Ubx (mouse monoclonal) | Developmental Studies Hybridoma | FP3.38 | IF(1:50) |
| Antibody | Anti-DsRed antibody (mouse monoclonal) | Santa Cruz Biotechnology | sc-390909 | IF(1:50) |
| Antibody | Anti-GFP antibody (rabbit polyclonal) | Thermo Fisher Scientific | A-11122 | IF(1:500) |
| Antibody | Anti-mouse Alexa Fluor 546 (goat polyclonal) | Thermo Fisher Scientific | A-11030 | IF(1:500) |
| Antibody | Anti-rabbit Alexa Fluor 488 (goat polyclonal) | Thermo Fisher Scientific | A-11008 | IF(1:500) |
| Chemical compound, drug | Paraformaldehyde 20% solution, EM Grade | Electron Microscopy Sciences | 15713S | |
| Chemical compound, drug | Formaldehyde, 16%, methanol free, Ultra Pure | Polysciences Inc | 18814-10 | |
| Chemical compound, drug | Phosphate-buffered saline (10×) pH 7.4, RNase-free | Thermo Fisher | AM9624 | |
| Chemical compound, drug | Tween 20 | Sigma | P1379 | |
| Chemical compound, drug | Triton X-100 | Bio-Rad | 161-0407 | |
| Chemical compound, drug | Tris base | Sigma | 11814273001 | |

*Appendix 1 Continued on next page*

*Appendix 1 Continued*

| Reagent type (species) or resource | Designation | Source or reference | Identifiers | Additional information |
|---|---|---|---|---|
| Chemical compound, drug | Methanol | Fisher Chemical | 203403 | |
| Chemical compound, drug | SSC, 20× | Thermo Fisher Scientific | 15557044 | |
| Chemical compound, drug | Formamide | Thermo Fisher Scientific | 17899 | |
| Chemical compound, drug | Dextran sulfate | Sigma | D8906 | |
| Chemical compound, drug | Salmon Sperm DNA | Thermo Fisher Scientific | AM9680 | |
| Chemical compound, drug | Ribonucleoside Vanadyl Complex | NEB | S1402S | |
| Chemical compound, drug | Nuclease-free BSA | Sigma | 126609 | |
| Chemical compound, drug | Triethylammonium acetate | Sigma | 625718 | |
| Chemical compound, drug | dGTP (100 MM) | VWR | 76510-208 | |
| Chemical compound, drug | dTTP (100 MM) | VWR | 76510-224 | |
| Chemical compound, drug | Lonza NuSieve 3:1 Agarose | Thermo Fisher Scientific | BMA50090 | |
| Chemical compound, drug | T4 DNA ligase | NEB | M0202L | |
| Chemical compound, drug | Biotin-11-dCTP | Jen Bioscience | NU-809-BIOX | |
| Chemical compound, drug | Biotin-14-dATP | Jen Bioscience | NU-835-BIO14 | |
| Chemical compound, drug | Qubit dsDNA HS Assay Kit | Life Technologies Corporation | Q32851 | |
| Chemical compound, drug | Phase Lock Gel, QuantaBio - 2302830, Phase Lock Gel Heavy | VMR | 10847-802 | |
| Chemical compound, drug | NEBNext Ultra II DNA Library Prep Kit for Illumina | NEB | E7645S | |
| Chemical compound, drug | Ampure Xp 5 ml Kit | Thermo Fisher Scientific | NC9959336 | |
| Chemical compound, drug | Hifi Hotstart Ready Mix | Thermo Fisher Scientific | 501965217 | |
| Chemical compound, drug | Dynabeads MyOne Streptavidin C1 | Life Technologies Corporation | 65001 | |
| Chemical compound, drug | cOmplete, EDTA-free Protease Inhibitor Cocktail | Sigma | 11873580001 | |
| Chemical compound, drug | N,N-Dimethylformamide | Sigma | 227056 | |

*Appendix 1 Continued on next page*

*Appendix 1 Continued*

| Reagent type (species) or resource | Designation | Source or reference | Identifiers | Additional information |
|---|---|---|---|---|
| Chemical compound, drug | Potassium acetate solution | Sigma | 95843 | |
| Chemical compound, drug | DSG (disuccinimidyl glutarate) | Thermo Fisher Scientific | PI20593 | |
| Chemical compound, drug | T4 Polynucleotide Kinase – 500 units | NEB | M0201S | |
| Chemical compound, drug | DNA Polymerase I, Large (Klenow) Fragment – 1000 units | NEB | M0210L | |
| Chemical compound, drug | End-it DNA End Repair Kit | Thermo Fisher Scientific | NC0105678 | |
| Chemical compound, drug | Proteinase K recomb. 100 mg | Sigma | 3115879001 | |
| Chemical compound, drug | Nuclease Micrococcal (s7) | Thermo Fisher Scientific | NC9391488 | |
| Chemical compound, drug | EGS (ethylene glycol bis(succinimidyl succinate)) | Thermo Fisher Scientific | PI21565 | |
| Sequence-based reagent | Fub-1 sense probe set | Biosearch Technologies | N/A | |
| Sequence-based reagent | Fub-1 antisense probe set | Biosearch Technologies | N/A | |
| Software, algorithm | Fiji (ImageJ) | *Schindelin et al., 2012* | | fiji.sc |
| Software, algorithm | NIS element | Nikon | | https://www.microscope.healthcare.nikon.com/products/software/nis-elements/ |
| Software, algorithm | GraphPad Prism 8 | GraphPad Software | | https://www.graphpad.com |
| Software, algorithm | HiGlass | *Kerpedjiev et al., 2018* | | http://higlass.io/ |
| Software, algorithm | bwa | *Li and Durbin, 2009* | | https://bio-bwa.sourceforge.net/ |
| Software, algorithm | samtools | Github/open source | | https://samtools.github.io |
| Software, algorithm | pairsamtools | Github/open source; *Goloborodko et al., 2023* | | https://github.com/open2c/pairtools |
| Software, algorithm | pairix | Github/open source; *Lee et al., 2021* | | https://github.com/4dn-dcic/pairix |
| Software, algorithm | cooler | *Abdennur et al., 2023* | | https://github.com/open2c/cooler |
| Software, algorithm | Miniconda | Anaconda | | https://docs.conda.io/en/latest/miniconda.html |
| Software, algorithm | Snakemake | Github/open source | | https://snakemake.github.io |

