## [Editor Report]

This study provides compelling evidence for the mechanism of an insulator element, which establishes boundaries between gene neighborhoods to allow proper gene regulation. In the fruit fly *Drosophila melanogaster*, the bithorax complex contains a series of Hox genes that determines segment identity. The authors show that transcriptional read-through of an bithorax insulator controls its activity and is used for proper patterning of the embryo.

---

## [Decision Letter]

**Decision letter after peer review:**

Thank you for submitting your article "The insulating activity of the *Drosophila* BX-C chromatin boundary Fub-1 is parasegmentally regulated by lncRNA read-through" for consideration by *eLife*. Your article has been reviewed by 2 peer reviewers, and the evaluation has been overseen by a Reviewing Editor and Kevin Struhl as the Senior Editor. The following individual involved in the review of your submission has agreed to reveal their identity: Ian Duncan (Reviewer #1).

First let me apologize for the long delay, which was due to administrative mixups, not science. The reviewers have discussed their reviews with one another, and the Reviewing Editor has drafted this to help you prepare a revised submission.

Essential revisions:

1) Given the central importance of the F1HS2 RNA to the story presented, it is disappointing that the structure and expression of this RNA are not well defined in the paper. In particular evidence of ncRNA transcript spanning HS1 in WT and absence in promoter inversion and premature termination strains would be important for the authors' model.

2) Please provide detail on the larval and imaginal phenotypes of HS1248 HS2505R animals.

3) Please address whether the direction of transcription of HS1 is important.

*Reviewer #1 (Recommendations for the authors):*

Page 5: The following sentences present what most readers will see as a contradiction:

"However, when Fab-7 is replaced by generic fly boundaries (e.g. scs or su(Hw)) or by BX-C boundaries that lack bypass activity (Mcp) these boundaries block crosstalk but do not support bypass. As a consequence, Abd-B expression in PS11/A6 is driven by iab-5 not iab-6 and this results in a loss-of-function phenotype".

How is it that iab-5 can drive Abd-B expression without bypassing the introduced generic boundaries? Others have attributed the ability of iab-5 to promote A5 identity in this situation to its ability to regulate abd-A.

Page 5: "where multiple boundaries are located between the iab-5 and iab-6 regulatory domain" should read "where multiple boundaries are located between iab-5 and the Abd-B promoter" or something similar.

Figure 1: Given the centrality of the bxd and F1HS2 RNAs to the story, it would help the reader if their locations were indicated in Figure 1.

Page 7: It would help the novice reader if at the beginning of the Results or in the Introduction it were pointed out that the Ubx domain is delimited and subdivided by three boundaries called Fub-2, Fub-1, and Fub. Without this statement, some readers will, at least initially, confuse these boundaries with one another or think they are synonymous.

Page 7 and General comment: Given the central importance of the F1HS2 RNA to the story presented, it is disappointing that the structure and expression of this RNA are not well defined in the paper. Although the characterization of F1HS2 is complicated by similar expression of the bxd RNA, the use of an existing strain in which the bxd promoter has been inverted could simplify this problem.

Figure 2 legend: State here what F1attP is, what LDIC means, and that the data are from whole 12-18hr embryos.

Page 10: "It would appear that this topological configuration is sufficient to block crosstalk between initiators in abx/bx and bxd/pbx when the activity state of these domains in T3 (PS5) is set during early embryogenesis." The logic here is difficult to follow. Either the argument should be expanded, or the sentence softened by substituting "It may be" for "It would appear that".

Figure 4 and Supplementary Figure 4: In wild-type males, the A6 tergite differs from the A5 tergite in having large regions devoid of trichomes. In Supplementary Figure 4 it looks to me that genotypes in which HS1 elements without HS2 are inserted into F7attP50 all show regions in A6 that are devoid of trichomes, indicating some degree of A6 identity. However, the images are not entirely clear. It is very important to resolve whether patches devoid of trichomes are indeed present in the A6 tergite, because, if so, these patches would indicate that iab-6 is able to bypass the HS1 insulator to some extent. These same HS1 insertions into F7attP50 present a second problem for the reader: A5 and A6 are pigmented, indicating that iab-5 is also able to bypass the HS1 insulator. In the absence of clarification here, many readers will be mystified and discouraged from trying to make sense of what is a rather complicated paper.

Figure 5: Contrary to the text, in part C, ventral nerve cord expression of HS2505R-HS1248R-SV40 and HS2505R-SV40-HS1248R look very similar to one another and to HS2505R-HS1248R. In addition, Abd-B expression in the ventral nerve cord should be shown for the HS2505R-mSV40-HS1248R genotype.

In part E, the right-hand genotype I believe should be HS2505R-HS1248R.

Page 20: Why reference Figure 2B here? Shouldn't this be 5E? The proper reference for the phenotypes of bxd and pbx mutations is definitely not Bender et al. – it should be one of the older Ed Lewis references!!!

Page 23: "activate Ubx expression in PS5/A1" should be "activate expression in PS6/A1". Also "is abrogated in PS6/A1 and more anterior parasegments/segments" should be "abrogated in PS6/A1 and more posterior parasegments/segments".

Page 26 top: "In the Fub-1 deletion, the proximal endpoint of the bxd/pbx TAD shifts to the Ubx promoter region". I don't see this shift in Figure 2 or in the schematic in Figure 7. Explain what is meant here or delete. In addition, "There could be several reasons for this discrepancy" is quite unclear. What discrepancy? Is it that one might expect the WT and F1attP data to look the same? The writing badly needs clarification here.

Page 27: "In this case, iab-6 would drive HS2 expression in PS11(A7) cells" should be "In this case, iab-6 would drive HS2 expression in PS11(A6) cells".

*Reviewer #2 (Recommendations for the authors):*

Overall, the work was well planned and conducted, and the results are clearly presented and interpreted. As mentioned in the public review, my main question is how general and significant the mechanism is in insulator function and animal development. Below are specific points.

1. Active transcription and chromatin boundaries are often found in close proximity. Can the authors check their micro-C and available PolII ChIP data from wt embryos to globally assess other known/novel boundary elements that show similar organization: an insulator (based on the presence of CTCF or other insulator protein binding sites) is near an active promoter that can transcribe through the insulator. In this way, can they predict other loci that may use a similar strategy in insulator regulation?

2. In Figures 4 and 5, the authors nicely showed that depleting HS2, adding the SV40 terminator, and changing the direction of the promoter all restore the insulator activity of HS1. Although these manipulations are supposed to remove or disrupt the transcriptional read-through of the non-coding transcript, the authors should validate the assumption. The authors can perform PolII ChIP-qPCR on some of the transgenes to directly show that the read-through is gone, for example in the HS2505R-SV40-HS1248R and the HS1248-HS2505R animals (and HS2505R- HS1248R) as a control.

3. Many promoters (especially for non-coding transcripts) are divergent, which means producing transcripts in both directions. However, the directionality of the HS2 promoter seems very strong, and maintaining such directionality is important for the function of HS2. Can the authors comment on what may ensure the directionality of the HS2 promoter?

4. There are a few typos in the text, and some labeling mistakes for the figures (line 398, shouldn't it be HS1248-HS2505R? line 492, figure 2; line 480, figure 2B, Figure 5E, the label for the second image).

---

## [Author Response]

Essential revisions:1) Given the central importance of the F1HS2 RNA to the story presented, it is disappointing that the structure and expression of this RNA are not well defined in the paper. In particular evidence of ncRNA transcript spanning HS1 in WT and absence in promoter inversion and premature termination strains would be important for the authors' model.

We have analyzed the “sense” lncRNA transcripts in WT and the Fub-1 deletion and anti-sense transcripts in WT, and the two HS2 promoter inversions. From a comparison of the smFISH hybridization pattern in WT and the Fub-1 deletion it is clear that both the *bxd* and *F1H2* transcripts are expressed in WT at the blastoderm stage. As the embryo develops, the *bxd* transcript disappears while the *F1H2* transcript is still detected. For the anti-sense transcripts—we didn’t detect them in WT, but they were observed in both of inverted replacements.

2) Please provide detail on the larval and imaginal phenotypes of HS1248 HS2505R animals.

Photos showing larval phenotype are provided.

3) Please address whether the direction of transcription of HS1 is important.

We didn’t do an experiment testing whether interruption of boundary activity required a “sense” transcript. However, some other experiments suggest that the transcript itself is unlikely to be relevant. We replaced *Fub-1* with two other BX-C boundaries, *Mcp* and *Fab-8*. Both block *bxd/pbx* from activating *Ubx* expression. However, when a P-element promoter is introduced “upstream” of *Mcp* or *Fab-8* (so it is pointing towards the boundaries and the *bxd/pbx* domain) it rescues the LOF phenotype. This would argue that the readthrough rather a particular feature of readthrough transcript is responsible for disrupting boundary activity.

Reviewer #1 (Recommendations for the authors):Page 5: The following sentences present what most readers will see as a contradiction: "However, when Fab-7 is replaced by generic fly boundaries (e.g. scs or su(Hw)) or by BX-C boundaries that lack bypass activity (Mcp) these boundaries block crosstalk but do not support bypass. As a consequence, Abd-B expression in PS11/A6 is driven by iab-5 not iab-6 and this results in a loss-of-function phenotype". How is it that iab-5 can drive Abd-B expression without bypassing the introduced generic boundaries? Others have attributed the ability of iab-5 to promote A5 identity in this situation to its ability to regulate abd-A.

Based on other experiments that we’ve done, it seems unlikely that *iab-5* generates an A5 segment via activation of *abd-A*. For example, other boundaries (e.g. Pita^x5^) are more effective in blocking iab-5 from activating *Abd-B* and give some LOF phenotypes in A5.

Page 5: "where multiple boundaries are located between the iab-5 and iab-6 regulatory domain" should read "where multiple boundaries are located between iab-5 and the Abd-B promoter" or something similar.

Typo – fixed.

Figure 1: Given the centrality of the bxd and F1HS2 RNAs to the story, it would help the reader if their locations were indicated in Figure 1.

Figure 1 is already very crowded. We added a schematic for bxd and Fub-1 ncRNAs in figure 7, right next to smFISH results.

Page 7: It would help the novice reader if at the beginning of the Results or in the Introduction it were pointed out that the Ubx domain is delimited and subdivided by three boundaries called Fub-2, Fub-1, and Fub. Without this statement, some readers will, at least initially, confuse these boundaries with one another or think they are synonymous.

This statement was added in introduction.

Page 7 and General comment: Given the central importance of the F1HS2 RNA to the story presented, it is disappointing that the structure and expression of this RNA are not well defined in the paper. Although the characterization of F1HS2 is complicated by similar expression of the bxd RNA, the use of an existing strain in which the bxd promoter has been inverted could simplify this problem.

We have analyzed lncRNA transcripts in WT, the *Fub-1* deletion and the two HS2 inversion. Our analysis of the lncRNA transcripts in pre-cellular blastoderm embryos in WT and the *Fub-1* using a probe adjacent to the *Fub-1* deletion indicates that both *bxd* and *F1HS2* transcripts are expressed at this stage. As the embryo develops the signal in the *Fub-1* deletion disappears while it is readily detected in WT. By stage 11, only WT has transcripts complementary to this probe.

We also looked at antisense transcripts adjacent to the *Fub-1* deletion in WT and the inverse *Fub-1* replacements. Antisense transcripts aren’t detected in WT but are in both of the inverse *Fub-1* replacements.

Figure 2 legend: State here what F1attP is, what LDIC means, and that the data are from whole 12-18hr embryos.

We have rewritten Figure 2 legend.

Page 10: "It would appear that this topological configuration is sufficient to block crosstalk between initiators in abx/bx and bxd/pbx when the activity state of these domains in T3 (PS5) is set during early embryogenesis." The logic here is difficult to follow. Either the argument should be expanded, or the sentence softened by substituting "It may be" for "It would appear that".

The sentence is rewritten.

Figure 4 and Supplementary Figure 4: In wild-type males, the A6 tergite differs from the A5 tergite in having large regions devoid of trichomes. In Supplementary Figure 4 it looks to me that genotypes in which HS1 elements without HS2 are inserted into F7attP50 all show regions in A6 that are devoid of trichomes, indicating some degree of A6 identity. However, the images are not entirely clear. It is very important to resolve whether patches devoid of trichomes are indeed present in the A6 tergite, because, if so, these patches would indicate that iab-6 is able to bypass the HS1 insulator to some extent. These same HS1 insertions into F7attP50 present a second problem for the reader: A5 and A6 are pigmented, indicating that iab-5 is also able to bypass the HS1 insulator. In the absence of clarification here, many readers will be mystified and discouraged from trying to make sense of what is a rather complicated paper.

In WT the trichome lawn in A5 is not always uniform and we often observe small patches of naked cuticle. So it wouldn’t be surprising to see small patches of naked cuticle in A6 in replacements that lack bypass activity. On the other hand, blocking crosstalk between *iab-6* and *iab-7* during the initiation phase—and rescuing the GOF transformation of A6A7—is likely different than blocking *iab-6* enhancers from activating *Abd-B* in PS11/A6. That this is likely the case in this instance is suggested by the sternite phenotype. While there are bristles on the A6 sternite (which shouldn’t be there) the morphology isn’t identical to A5. Instead of a quadrilateral shape, the sternite still has hints of the “banana” shape. This is also true when *Fab-7* is replaced by a 209 bp fragment from *Fab-8* that has blocking activity but lacks full bypass activity—which is conferred by a neighboring 165 bp fragment. In other replacements, such *Pita^x5^* the A6 sternite looks like A5. In the *Pita^x5^* replacement, the trichome pattern in A5 resembles A4 in being more densely packed and the lack of small patches of naked cuticle.

Figure 5: Contrary to the text, in part C, ventral nerve cord expression of HS2505R-HS1248R-SV40 and HS2505R-SV40-HS1248R look very similar to one another and to HS2505R-HS1248R. In addition, Abd-B expression in the ventral nerve cord should be shown for the HS2505R-mSV40-HS1248R genotype.

We agree that difference in expression between these strains is hard to see with naked eye. We added supplementary Figure 5—figure supplement 1 that shows fluorescence intensity plot profiles of *HS2^505R^-HS1^248R^, HS2^505R^-HS1^248R^-SV40* and *HS2^505R^-SV40*-*HS1^248R^* embryos. Figure 5—figure supplement 1 shows that the intensity of the signal is lower in PS11/A6 in *HS2^505R^-SV40*-*HS1^248R^* compared to both *HS2^505R^-HS1^248R^* and *HS2^505R^-HS1^248R^-SV40.*

In part E, the right-hand genotype I believe should be HS2505R-HS1248R.

Error is fixed.

Page 20: Why reference Figure 2B here? Shouldn't this be 5E? The proper reference for the phenotypes of bxd and pbx mutations is definitely not Bender et al. – it should be one of the older Ed Lewis references!!!

The reference is fixed, the proper citations have been added.

Page 23: "activate Ubx expression in PS5/A1" should be "activate expression in PS6/A1". Also "is abrogated in PS6/A1 and more anterior parasegments/segments" should be "abrogated in PS6/A1 and more posterior parasegments/segments".

The typos are fixed.

Page 26 top: "In the Fub-1 deletion, the proximal endpoint of the bxd/pbx TAD shifts to the Ubx promoter region". I don't see this shift in Figure 2 or in the schematic in Figure 7. Explain what is meant here or delete.

We refer to a subTAD formed by Fub-1 and bxd PRE. Clarification was added to the text.

In addition, "There could be several reasons for this discrepancy" is quite unclear. What discrepancy? Is it that one might expect the WT and F1attP data to look the same? The writing badly needs clarification here.

The point we are trying to make is that a TAD-like MicroC pattern is observed in WT—even though boundary function should, in principle, be disrupted in ~2/3rds of the cells in the embryo. The TAD-like signal we observed could be from the remaining 1/3^rd^ of the embryos. On the other hand, (a) not all of the cells in the posterior have detectable F1HS2 transcripts and (b) transcription is not continuous—but instead comes in bursts. Thus, additional explanation is that the disruption of boundary function is also not continuous, but rather linked to the bursts. We hope this point is now clear.

Page 27: "In this case, iab-6 would drive HS2 expression in PS11(A7) cells" should be "In this case, iab-6 would drive HS2 expression in PS11(A6) cells".

Typo is fixed.

Reviewer #2 (Recommendations for the authors):Overall, the work was well planned and conducted, and the results are clearly presented and interpreted. As mentioned in the public review, my main question is how general and significant the mechanism is in insulator function and animal development. Below are specific points.1. Active transcription and chromatin boundaries are often found in close proximity. Can the authors check their micro-C and available PolII ChIP data from wt embryos to globally assess other known/novel boundary elements that show similar organization: an insulator (based on the presence of CTCF or other insulator protein binding sites) is near an active promoter that can transcribe through the insulator. In this way, can they predict other loci that may use a similar strategy in insulator regulation?

This is a good idea and we are planning to do this type of analysis. Probably the most interesting possibility is that these events will be tissue/cell type specific—which is what we see here. This will complicate the analysis in flies (eg., a CNS specific regulation) unless one can isolate specific tissues/cell types and do both microC and RNA-seq.

2. In Figures 4 and 5, the authors nicely showed that depleting HS2, adding the SV40 terminator, and changing the direction of the promoter all restore the insulator activity of HS1. Although these manipulations are supposed to remove or disrupt the transcriptional read-through of the non-coding transcript, the authors should validate the assumption. The authors can perform PolII ChIP-qPCR on some of the transgenes to directly show that the read-through is gone, for example in the HS2505R-SV40-HS1248R and the HS1248-HS2505R animals (and HS2505R- HS1248R) as a control.

We are planning to conduct this study using transgenes.

3. Many promoters (especially for non-coding transcripts) are divergent, which means producing transcripts in both directions. However, the directionality of the HS2 promoter seems very strong, and maintaining such directionality is important for the function of HS2. Can the authors comment on what may ensure the directionality of the HS2 promoter?

We didn’t detect transcripts from HS2 in WT going the opposite direction. However, the reason could be that transcription in the “forward” direction occurs at a much higher frequency than in the “reverse” orientation. So if you are looking by deep sequencing you can detect the reverse transcripts, while they are no readily detected using in situ hybridization.

4. There are a few typos in the text, and some labeling mistakes for the figures (line 398, shouldn't it be HS1248-HS2505R? line 492, figure 2; line 480, figure 2B, Figure 5E, the label for the second image).

All typos are fixed.